# Towards a Statistical Understanding of Neural Networks: Beyond the Neural Tangent Kernel Theories

## Abstract

A primary advantage of neural networks lies in their feature learning characteristics, which is challenging to theoretically analyze due to the complexity of their training dynamics. We examine feature learning and its potential benefits for generalization from a statistical perspective. After reviewing the neural tangent kernel (NTK) theory and recent results in kernel regression, which address the generalization issue of sufficiently wide neural networks, we examine limitations and implications of the fixed kernel theory (as the NTK theory) and review recent theoretical advancements in feature learning. Moving beyond theories with fixed features, we consider neural networks as adaptive feature models. Finally, we propose an over-parameterized Gaussian sequence model as a prototype for the adaptive feature model to study feature learning characteristics and motivate their future analysis for neural networks.

**Keywords:** neural networks, generalization ability, feature learning, kernel regression, over-parameterized Gaussian sequence model

**Mathematics Subject Classification (2020):** 62G05

## 1 Introduction

In recent years, deep neural networks have made remarkable accomplishments in many application areas, whereas their theoretical understanding has lagged far behind. Although neural network modeling has a long history, recent developments in data availability, computing resources, and network architecture designs were believed to be the key to their somewhat mysteriously outstanding performance.

Numerous topics have been explored regarding neural network theory, such as those discussed in recent review papers and references therein (Bartlett et al., 2021; Belkin, 2021; Fan et al., 2021; Suh and Cheng, 2024). In this paper, we consider the generalization ability of neural networks within the nonparametric regression framework. Let a compact set $\mathcal{X} \subset \mathbb{R}^d$ be the input space, let $\mathcal{Y} \subseteq \mathbb{R}$ be the output space, and let $\mu$ be the input distribution on $\mathcal{X}$. Suppose that the $n$ i.i.d. draws $\{(\boldsymbol{x}_i, y_i)\}_{i=1}^n$ are sampled from the model

$$y = f^*(\boldsymbol{x}) + \epsilon, \quad \boldsymbol{x} \sim \mu, \tag{1}$$

where $f^* \in L^2(\mathcal{X}, \mu)$ is the true function, and the noise $\epsilon$, independent of $\boldsymbol{x}$, has zero mean and variance $\sigma_0^2 > 0$. Throughout the paper, we denote $\boldsymbol{X} = (\boldsymbol{x}_1, \ldots, \boldsymbol{x}_n) \in \mathbb{R}^{d \times n}$, and $\boldsymbol{Y} = (y_1, \ldots, y_n)^\top$. For any estimator $\hat{f}$ (which can be generated by a neural network, a kernel

regression model, etc.), we will investigate the $L^2$-norm generalization error (or excess risk) given by

$$\mathcal{R}(\hat{f}) = \left\|\hat{f} - f^*\right\|_2^2 \triangleq \mathbb{E}_{\boldsymbol{x} \sim \mu}\left[\left(\hat{f}(\boldsymbol{x}) - f^*(\boldsymbol{x})\right)^2\right]. \tag{2}$$

It is natural to ask the following questions about neural networks:

(i) How can the generalization ability of neural networks be characterized from a statistical viewpoint? For instance, can we establish convergence rates of the generalization error with respect to sample sizes? Do we know the minimax optimality of a neural network method?

(ii) Why do neural networks outperform other existing methods in many applications? Can we understand when and how?

To answer the first question, the line of work of Bauer and Kohler (2019), Schmidt-Hieber (2020), and Suzuki (2018) considered the "algorithm-independent controls" of the generalization error, a name coined by Fan et al. (2021). By carefully selecting the candidate function class represented by a specific neural network architecture, they obtained generalization error bounds for the empirical risk minimization estimator over the chosen function class. However, the empirical risk minimization estimator is not necessarily the output of a specified training procedure. For a parameterized neural network $f_{\boldsymbol{\theta}}$, training acts on the empirical risk

$$\mathcal{L}_n(\boldsymbol{\theta}) = \frac{1}{n}\sum_{i=1}^{n}\left(y_i - f_{\boldsymbol{\theta}}(\boldsymbol{x}_i)\right)^2. \tag{3}$$

The map $\boldsymbol{\theta} \mapsto \mathcal{L}_n(\boldsymbol{\theta})$ is generally nonconvex. Thus, an estimator-level guarantee does not by itself show that a specified training procedure reaches a parameter producing that estimator. Initialization and training choices can lead to different parameter trajectories and predictors.

For comparison, our current paper focuses on *algorithm-dependent controls* (Fan et al., 2021), which take into account the training dynamics of neural networks. In practice, training dynamics are usually difficult to understand and can lead to drastically different generalization behaviors. This makes the theoretical analysis of training dynamics one of the most intriguing and challenging tasks.

Our starting point is the seminal *neural tangent kernel* (NTK) theory (Arora et al., 2019b; Jacot et al., 2018; Lee et al., 2019), which establishes a connection between the training dynamics of sufficiently wide neural networks and the kernel gradient flow (Section 2.1). Due to the success of the NTK theory, we have experienced a renaissance of kernel regression. We will review some important topics about the generalization ability of kernel regression, addressing both fixed-dimensional (Section 2.3) and high-dimensional settings (Section 2.4).

The answer to the second question is even more complicated. It is believed that the superiority of neural networks stems from their feature learning characteristics, which cannot be addressed by the NTK theory. Recent work has examined finite-width NTK behavior, the scope and limitations of NTK theory, and the evolution of NTK eigenvectors at the edge of stability (Golikov et al., 2022; Jiang et al., 2025; Seleznova and Kutyniok, 2022). Therefore, another theme of this paper is to go beyond the NTK theory and study feature learning characteristics of neural networks. Specifically, in Section 3, we discuss limitations and implications of the NTK

theory (as the fixed kernel regression) and review some recent advances in feature learning theory. In Section 4, we consider an *adaptive feature model* to emulate the feature learning process of neural networks. Furthermore, we introduce the *over-parameterized Gaussian sequence model* as a viable prototype for the adaptive feature model. Lecué et al. (2025) formalize the distinction between alignment for a fixed representation and feature learning, whereas Gavrilopoulos et al. (2024) train a representation in the first layer and then analyze kernel ridge regression with the resulting conjugate kernel learned from the data, including an excess risk result for a single neuron example. Relative to these works, our contribution is the Gaussian sequence prototype with learnable rotations and rescalings, rather than a new general definition of feature learning, a training mechanism for neural networks, or a mathematical guarantee. This model offers a tractable way to formulate questions about training dynamics and feature learning. Finally, we discuss some future questions in Section 5. We hope that our review and the proposed prototype can offer new insights and directions for developing theoretical understandings of neural networks.

**Notations**: The asymptotic notations, $O(\cdot)$, $o(\cdot)$, $\Omega(\cdot)$ and $\Theta(\cdot)$, will be used throughout the paper: We say $a_n = \Omega(b_n)$ if $b_n = O(a_n)$; $a_n = \Theta(b_n)$ if $a_n = O(b_n)$ and $a_n = \Omega(b_n)$. We also write $a_n \asymp b_n$ for $a_n = \Theta(b_n)$, and $a_n \lesssim b_n$ for $a_n = O(b_n)$. The corresponding probabilistic versions of the asymptotic notations will also be used: $O_{\mathbb{P}}(\cdot), o_{\mathbb{P}}(\cdot), \Omega_{\mathbb{P}}(\cdot)$ and $\Theta_{\mathbb{P}}(\cdot)$. For example, we say that random variables $X_n, Y_n$ satisfy $X_n = O_{\mathbb{P}}(Y_n)$ if $\forall\ \varepsilon > 0$, there exist constants $C_\varepsilon$ and $N_\varepsilon$ such that $\mathbb{P}\left(|X_n| \geq C_\varepsilon |Y_n|\right) \leq \varepsilon$ for $n > N_\varepsilon$. For $\alpha > 0$, we write $\lfloor \alpha \rfloor$ as the largest integer not exceeding $\alpha$. We use boldface letters to denote column vectors and matrices; $\|\boldsymbol{x}\|_2$ is the $\ell^2$-norm of vector $\boldsymbol{x}$.

## 2 Neural tangent kernel theory and kernel regression

### 2.1 Neural tangent kernel theory

The neural tangent kernel (NTK) theory (Arora et al., 2019b; Jacot et al., 2018; Lee et al., 2019) is one of the most successful tools for analyzing neural networks' training dynamics and generalization ability. A major challenge in studying neural networks is the highly non-convex nature of the target function. The basic idea of the NTK theory is that when a neural network's width is sufficiently large, the network's training dynamics can be approximated well by the kernel gradient flow for the corresponding neural tangent kernel, which is a convex problem. Numerous papers discuss various topics within the NTK theory in different settings, e.g. Lai et al. (2023), Du et al. (2019), Huang et al. (2020), Arora et al. (2019a), Li et al. (2024c), and Nitanda and Suzuki (2020).

A purpose of our introduction of the NTK theory is to emphasize the importance of studying kernel regression, which prepares us for proposing the adaptive feature approach in subsequent sections. Therefore, here we only review one result from Lai et al. (2023) to establish the connection between neural networks and kernel gradient flow. Specifically, Lai et al. (2023) considered the following two-layer fully connected ReLU neural network with width $m$:

$$f_{\boldsymbol{\theta}}^m(\boldsymbol{x}) = \frac{1}{\sqrt{m}} \sum_{r=1}^{m} a_r \sigma\left(\boldsymbol{w}_r^\top \boldsymbol{x} + b_r\right) + b, \tag{4}$$

where $\boldsymbol{\theta} = \text{vec}(\{a_r, \boldsymbol{w}_r, b_r, b, r = 1, \ldots, m\})$ and $\sigma(\cdot)$ is the ReLU function. The neural network is trained through the gradient flow of parameters $\boldsymbol{\theta}(t)$ to minimize the mean-squared loss:

$$\mathcal{L}(f_{\boldsymbol{\theta}}^m) = \frac{1}{2n} \sum_{i=1}^{n} (y_i - f_{\boldsymbol{\theta}}^m(\boldsymbol{x}_i))^2,$$

i.e.,

$$\frac{\mathrm{d}}{\mathrm{d}t} \boldsymbol{\theta}(t) = -\nabla_{\boldsymbol{\theta}} \mathcal{L}\left(f_{\boldsymbol{\theta}(t)}^m\right).$$

Then, $f_{\boldsymbol{\theta}(t)}^m$ is the neural network estimator at time $t$ (see Lai et al. 2023, Section 3 for more details). The first-order linearization of $f_{\boldsymbol{\theta}}^m$ around $\boldsymbol{\theta}(0)$ has tangent features $\nabla_{\boldsymbol{\theta}} f_{\boldsymbol{\theta}(0)}^m(\boldsymbol{x})$, whose inner products define the corresponding NTK:

$$k_{\text{NTK}}(\boldsymbol{x}, \boldsymbol{x}') = \left\langle \nabla_{\boldsymbol{\theta}} f_{\boldsymbol{\theta}(0)}^m(\boldsymbol{x}), \nabla_{\boldsymbol{\theta}} f_{\boldsymbol{\theta}(0)}^m(\boldsymbol{x}') \right\rangle.$$

Accordingly, denote $\hat{f}_t^{\text{NTK}}$ as the kernel gradient flow estimator (defined later in Definition 2.2.2) with this kernel. Under certain initialization, Lai et al. (2023), Proposition 2, showed that for any $\varepsilon > 0$, if the width $m$ is sufficiently large, then

$$\sup_{t \geq 0} \sup_{\boldsymbol{x} \in \mathcal{X}} \left| f_{\boldsymbol{\theta}(t)}^m(\boldsymbol{x}) - \hat{f}_t^{\text{NTK}}(\boldsymbol{x}) \right| \leq \varepsilon,$$

and

$$\sup_{t \geq 0} \left| \mathcal{R}\left(f_{\boldsymbol{\theta}(t)}^m\right) - \mathcal{R}\left(\hat{f}_t^{\text{NTK}}\right) \right| \leq \varepsilon,$$

hold with probability at least $1 - o_m(1)$, where the randomness comes from the initialization of the parameters. In other words, as the width $m$ tends to infinity, the neural network estimator $f_{\boldsymbol{\theta}(t)}^m$ uniformly converges to $\hat{f}_t^{\text{NTK}}$, and the corresponding generalization error $\mathcal{R}(f_{\boldsymbol{\theta}(t)}^m)$ is well approximated by $\mathcal{R}(\hat{f}_t^{\text{NTK}})$. Consequently, the generalization ability of the neural network estimator can be studied through the kernel gradient flow estimator. Similar results have also been derived for other neural network architectures (Arora et al., 2019b; Tirer et al., 2022, etc.), making the kernel gradient flow a reasonable alternative for sufficiently wide neural networks and leading to a resurgence in the study of kernel regression.

## 2.2 Preliminaries of kernel regression

In kernel regression (or kernel method), we are given a pre-specified kernel function $k(\cdot, \cdot) : \mathcal{X} \times \mathcal{X} \to \mathbb{R}$, which is supposed to be positive-definite, symmetric, and continuous. Without loss of generality, we assume that the kernel function is bounded by 1, that is, $\sup_{\boldsymbol{x} \in \mathcal{X}} k(\boldsymbol{x}, \boldsymbol{x}) \leq 1$. Then, there exists a corresponding function space $\mathcal{H} \subset L^2(\mathcal{X}, \mu)$, called the reproducing kernel Hilbert space (RKHS), with inner product $\langle \cdot, \cdot \rangle_{\mathcal{H}}$ and norm $\| \cdot \|_{\mathcal{H}}$. Mercer's theorem (see Steinwart and Christmann 2008, Theorem 4.49) shows that there exists a non-increasing summable sequence $\{\lambda_j\}_{j=1}^{\infty} \subset (0, \infty)$ and a family of functions $\{\psi_j\}_{j=1}^{\infty} \subset \mathcal{H}$, such that

$$k(\boldsymbol{x}, \boldsymbol{x}') = \sum_{j=1}^{\infty} \lambda_j \psi_j(\boldsymbol{x}) \psi_j(\boldsymbol{x}'), \quad \boldsymbol{x}, \boldsymbol{x}' \in \mathcal{X}, \tag{5}$$

where the convergence is absolute and uniform. $\{\psi_j\}_{j=1}^{\infty}$ and $\{\lambda_j\}_{j=1}^{\infty}$ are the eigenfunctions and eigenvalues, respectively, of the kernel function $k$ and the RKHS $\mathcal{H}$. The set of functions

$\{\psi_j\}_{j=1}^{\infty}$ can be assumed to be an orthonormal basis of $L^2(\mathcal{X}, \mu)$ without loss of generality. The RKHS $\mathcal{H}$ can be expressed as

$$\mathcal{H} = \left\{ \sum_{j=1}^{\infty} a_j \lambda_j^{\frac{1}{2}} \psi_j(\cdot) : \ (a_j)_{j=1}^{\infty} \in \ell^2 \right\},$$

where $\ell^2$ denotes the space of square-summable sequences. The corresponding norm is

$$\left\| \sum_{j=1}^{\infty} a_j \lambda_j^{\frac{1}{2}} \psi_j(\cdot) \right\|_{\mathcal{H}} = \left( \sum_{j=1}^{\infty} a_j^2 \right)^{\frac{1}{2}}.$$

For more details on RKHS, see Steinwart and Christmann (2008) and Steinwart and Scovel (2012).

The basic idea of kernel regression is to estimate $f^*$ using candidate functions from $\mathcal{H}$ (Cucker and Smale, 2001; Kohler and Krzyżak, 2001; Steinwart and Christmann, 2008). A large class of kernel regression estimators is collectively introduced as spectral algorithms (Caponnetto, 2006; Gerfo et al., 2008; Rosasco et al., 2005). The two most widely studied are the *kernel ridge regression* and *kernel gradient flow*. For a kernel function $k$, we denote

$$\mathbb{K}(\boldsymbol{X}, \boldsymbol{X}) = (k(\boldsymbol{x}_i, \boldsymbol{x}_j))_{n \times n}, \quad \mathbb{K}(\boldsymbol{x}, \boldsymbol{X}) = (k(\boldsymbol{x}, \boldsymbol{x}_1), \dots, k(\boldsymbol{x}, \boldsymbol{x}_n)).$$

**Definition 2.2.1** (Kernel ridge regression, KRR)**.** *For a given kernel function $k$, corresponding RKHS $\mathcal{H}$ and any $0 < \lambda < \infty$, kernel ridge regression constructs an estimator $\hat{f}_\lambda$ by solving the penalized least square problem*

$$\hat{f}_\lambda = \underset{f \in \mathcal{H}}{\arg\min} \left( \frac{1}{n} \sum_{i=1}^{n} (y_i - f(\boldsymbol{x}_i))^2 + \lambda \|f\|_{\mathcal{H}}^2 \right),$$

*where $\lambda$ is the regularization parameter. The explicit expression of $\hat{f}_\lambda$ is*

$$\hat{f}_\lambda(\boldsymbol{x}) = \mathbb{K}(\boldsymbol{x}, \boldsymbol{X}) \left( \mathbb{K}(\boldsymbol{X}, \boldsymbol{X}) + n\lambda \mathbf{I}_n \right)^{-1} \boldsymbol{Y}.$$

**Definition 2.2.2** (Kernel gradient flow, KGF)**.** *For a given kernel function $k$, corresponding RKHS $\mathcal{H}$, and any $0 < t < \infty$, kernel gradient flow constructs an estimator $\hat{f}_t$ by solving the differential equation*

$$\frac{\mathrm{d}}{\mathrm{d}t} \hat{f}_t(\boldsymbol{x}) = -\frac{1}{n} \mathbb{K}(\boldsymbol{x}, \boldsymbol{X}) \left( \hat{f}_t(\boldsymbol{X}) - \boldsymbol{Y} \right), \tag{6}$$

*and let $\hat{f}_0(\boldsymbol{x}) \equiv 0$. For any $t > 0$, when the matrix $\mathbb{K}(\boldsymbol{X}, \boldsymbol{X})$ is strictly positive definite, the explicit expression of $\hat{f}_t$ is*

$$\hat{f}_t(\boldsymbol{x}) = \mathbb{K}(\boldsymbol{x}, \boldsymbol{X}) \mathbb{K}(\boldsymbol{X}, \boldsymbol{X})^{-1} \left( \mathbf{I}_n - e^{-\frac{1}{n} \mathbb{K}(\boldsymbol{X}, \boldsymbol{X})t} \right) \boldsymbol{Y}. \tag{7}$$

For finite $t$, the differential equation (6) defines the kernel gradient flow estimator $\hat{f}_t \in \mathcal{H}$ minimizing the training loss $\|\hat{f}_t(\boldsymbol{X}) - \boldsymbol{Y}\|_2^2/n$. When $\mathbb{K}(\boldsymbol{X}, \boldsymbol{X})$ is strictly positive definite, only its limit as $t \to \infty$ yields the *kernel interpolation estimator*, which is the $\lambda \to 0$ limit of kernel ridge regression:

$$\hat{f}_{\mathrm{inter}}(\boldsymbol{x}) = \mathbb{K}(\boldsymbol{x}, \boldsymbol{X}) \mathbb{K}(\boldsymbol{X}, \boldsymbol{X})^{-1} \boldsymbol{Y}, \tag{8}$$

which will be of independent interest.

In the rate comparisons considered below, $t$ plays the same role as the regularization parameter $\lambda$ in KRR: when $t \asymp \lambda^{-1}$, their generalization rates coincide up to the saturation effect of KRR, which will be introduced later. In the definition of general spectral algorithms, both $t$ and $\lambda$ can be unified as a regularization parameter in the spectral algorithm's filter function (see, e.g., Zhang et al. 2024a, Definition 1). Additionally, the kernel gradient flow is a continuous version of the kernel gradient descent, which has similar theoretical properties and is used more frequently in practice.

In the following, we focus on kernel ridge regression and kernel gradient flow. For studies of general spectral algorithms, see Lin et al. (2018), Blanchard and Mücke (2018), Lin and Cevher (2020), Zhang et al. (2024a), and Li et al. (2024a).

## 2.3 Kernel regression in fixed dimensions

### 2.3.1 Assumptions

In fixed dimensions, we disregard the constants' dependence on the input dimension $d$. The first commonly used assumption is the eigenvalue decay rate of the eigenvalues $\{\lambda_j\}_{j=1}^{\infty}$.

**Assumption 1** (Eigenvalue decay rate, EDR). *Suppose that the eigenvalue decay rate (EDR) of $\mathcal{H}$ is $\beta > 1$. That is, there exist positive constants $c$ and $C$ such that*

$$cj^{-\beta} \leq \lambda_j \leq Cj^{-\beta}, \quad \forall j = 1, 2, \ldots.$$

Assumption 1 holds for many kernels, e.g., the Laplacian kernel, Matérn kernel, neural tangent kernel, etc., and is also closely related to the effective dimension or the capacity condition of RKHS (Caponnetto and de Vito, 2007).

The second widely adopted assumption is the source condition, which characterizes the relative smoothness of $f^*$ with respect to $\mathcal{H}$. To introduce this, we need the following definition of the *interpolation space* (or power space) $[\mathcal{H}]^s$ for any $s \geq 0$, which is defined as

$$[\mathcal{H}]^s = \left\{ \sum_{j=1}^{\infty} a_j \lambda_j^{\frac{s}{2}} \psi_j(\cdot) : (a_j)_{j=1}^{\infty} \in \ell^2 \right\}, \tag{9}$$

equipped with the norm

$$\left\| \sum_{j=1}^{\infty} a_j \lambda_j^{\frac{s}{2}} \psi_j(\cdot) \right\|_{[\mathcal{H}]^s} = \left( \sum_{j=1}^{\infty} a_j^2 \right)^{\frac{1}{2}}.$$

Specifically, we have $[\mathcal{H}]^1 = \mathcal{H}$. For $0 < s_1 < s_2$, the inclusion maps $[\mathcal{H}]^{s_2} \hookrightarrow [\mathcal{H}]^{s_1} \hookrightarrow [\mathcal{H}]^0$ are compact (Fischer and Steinwart, 2020); here $\hookrightarrow$ denotes a continuous embedding. The functions in $[\mathcal{H}]^s$ with smaller $s$ are less "smooth", which are harder for a kernel regression algorithm to estimate. In fact, another equivalent definition of $[\mathcal{H}]^s$ is through the *real interpolation* in functional analysis (Sawano, 2018; Steinwart and Scovel, 2012; Tartar, 2007).

**Assumption 2** (Source condition). *Suppose that for some $s \geq 0$, there is a constant $R > 0$ such that $f^* \in [\mathcal{H}]^s$ and*

$$\|f^*\|_{[\mathcal{H}]^s} \leq R.$$

*We refer to $s$ as the source condition of $f^*$.*

### 2.3.2 Learning curve results

The first question of interest is the minimax optimality of kernel regression. In the framework of eigenvalue decay rate and source conditions, the minimax optimality was first established in the well-specified case ($f^* \in \mathcal{H}$, or the source condition $s \geq 1$) in Caponnetto (2006); Caponnetto and de Vito (2007). Then, extensive subsequent literature (Celisse and Wahl, 2020; Dieuleveut and Bach, 2016; Fischer and Steinwart, 2020; Li et al., 2024b; Pillaud-Vivien et al., 2018; Steinwart et al., 2009; Wang and Jing, 2022; Zhang et al., 2023) studied mis-specified case (source condition $0 < s < 1$). Among them, Fischer and Steinwart (2020) firstly considered the embedding property of $\mathcal{H}$: we say that $\mathcal{H}$ has an embedding property of order $\alpha \in (0,1]$ if $[\mathcal{H}]^\alpha$ can be continuously embedded into $L^\infty(\mathcal{X}, \mu)$, i.e., the operator norm of the embedding satisfies

$$\left\| [\mathcal{H}]^\alpha \hookrightarrow L^\infty(\mathcal{X}, \mu) \right\|_{\mathrm{op}} = M_\alpha < \infty.$$

This embedding property was later summarized as an embedding index assumption (see, e.g., Li et al. 2024b, Assumption 2) and led to the minimax optimality of spectral algorithms for $0 < s < 1$ (Zhang et al., 2024a). The embedding index assumption postulates that $\alpha_0 = 1/\beta$, where $\alpha_0$ is defined as

$$\alpha_0 = \inf \left\{ \alpha \in [1/\beta, 1] : \left\| [\mathcal{H}]^\alpha \hookrightarrow L^\infty(\mathcal{X}, \mu) \right\|_{\mathrm{op}} < \infty \right\}.$$

The minimax optimality-type results considered the upper bound of the generalization error of an algorithm and the algorithm-independent minimax lower bound. In the renaissance of kernel regression arising from the study of neural networks, some recent work (Bordelon et al., 2020; Cui et al., 2021; Li et al., 2024a,b) further considered the learning curve of kernel regression, which aims to obtain precise formulas or exact order of the generalization error (both upper and lower bounds) under any choice of the regularization parameter and even any noise level. The learning curve-type results provide a nearly comprehensive picture of an estimator's generalization ability. Note that in order to obtain a reasonable lower bound in the learning curve scenario, we need to assume that the source condition of $f^*$ is exactly $s$, i.e., $f^* \in [\mathcal{H}]^s$, and $f^* \notin [\mathcal{H}]^r, \forall r > s$. Specific descriptions of this condition were provided in these learning curve papers (see Cui et al. 2021, Eq.(8); or Li et al. 2024b, Assumption 3).

Next, we formally state the learning curve result for kernel gradient flow (from Li et al. 2024a, Theorem 3.1). Under Assumptions 1 and 2, with $\beta > 1, s > 0$, and the embedding index assumption, by choosing $t \asymp n^\theta, \theta > 0$, we have

$$\mathcal{R}(\hat{f}_t) = \begin{cases} \Theta_{\mathbb{P}} \left( n^{-s\theta} + n^{-(1-\theta/\beta)} \right), & \text{if } \theta < \beta, \\ \sigma_0^2 \cdot \tilde{\Omega}_{\mathbb{P}}(1), & \text{if } \theta \geq \beta, \end{cases} \tag{10}$$

where $a_n = \tilde{\Omega}(b_n)$ means $a_n = \Omega((\ln n)^{-p} b_n)$ for any $p > 0$, and $\tilde{\Omega}_{\mathbb{P}}$ is the probability version of $\tilde{\Omega}$. In fact, Li et al. (2024a) addressed all analytic spectral algorithms, including KRR and KGF.

The first line of (10) is the regularized regime, where the two terms correspond to the bias and variance terms, respectively. As a direct corollary, choosing the optimal regularization $t_{\mathrm{opt}} \asymp n^{\frac{\beta}{s\beta+1}}$ leads to the optimal convergence rate of the kernel gradient flow:

$$n^{-\frac{s\beta}{s\beta+1}}, \tag{11}$$

which is also the minimax lower rate of the function space $[\mathcal{H}]^s$. The second line of (10) is the interpolating regime, where the regularization is slight, and the estimator behaves similarly to the kernel interpolation estimator (see Section 2.3.3). The result implies that the generalization ability in the interpolating regime can be arbitrarily bad. Figure 1 visualizes the convergence rates for different values of $\theta > 0 : t \asymp n^{\theta}$, and source condition $s > 0$. The dashed lines in Figure 1 (a) represent the minimax rates achieved by the optimal regularization $t$.

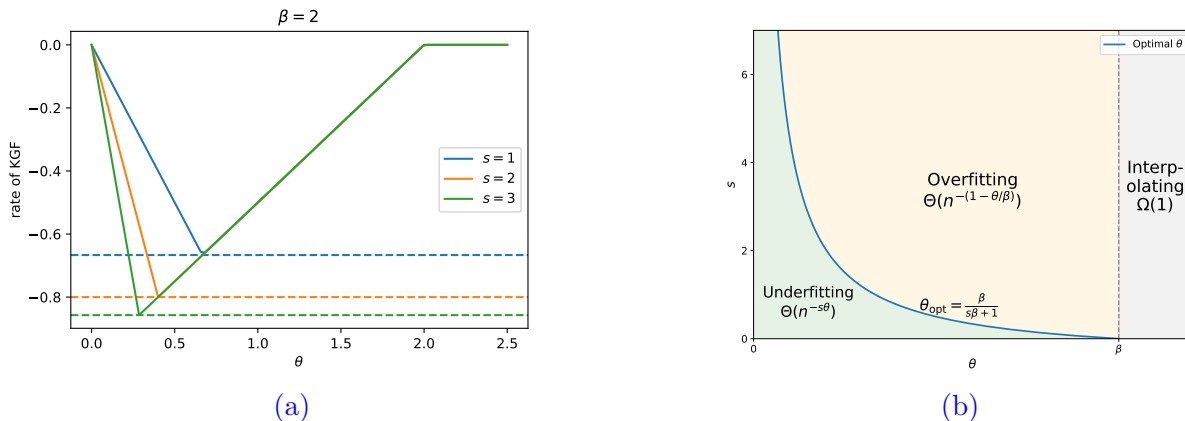

(a)                                             (b)

Figure 1: Asymptotic learning curve of kernel gradient flow.

Cui et al. (2021) and Li et al. (2024b) also discussed the learning curve of the kernel ridge regression and the impact of the noise level. Compared with the convergence rates of KGF in (10), the difference in KRR is the saturation effect, while KGF never saturates. The saturation effect, first conjectured by Bauer et al. (2007), says that when $s > 2$, regardless of how carefully the KRR is tuned, the convergence rate cannot be faster than $n^{-\frac{2\beta}{2\beta+1}}$, which is suboptimal compared with the minimax rate $n^{-\frac{s\beta}{s\beta+1}}$.

To end this subsection, we consider an example of the Sobolev RKHS. Assume that $\mathcal{X} \subset \mathbb{R}^d$ is a bounded domain with a smooth boundary and that the input distribution $\mu$ on $\mathcal{X}$ has Lebesgue density $p$ satisfying $0 < c \leq p(x) \leq C$ for two constants $c$ and $C$. The (fractional) Sobolev space $W^{m,2}(\mathcal{X})$ is an RKHS if $m > d/2$ (Adams and Fournier, 2003). Denoting $\mathcal{H} = W^{m,2}(\mathcal{X})$, previous results have shown that the decay rate of the eigenvalues of $\mathcal{H}$ is $\beta = 2m/d$ (Edmunds and Triebel, 1996), and $\mathcal{H}$ satisfies the embedding index assumption $\alpha_0 = 1/\beta$. Furthermore, the interpolation space of $\mathcal{H}$ is still a Sobolev space, i.e., $[\mathcal{H}]^p \cong W^{mp,2}(\mathcal{X}), \forall p > 0$. Suppose that the true function $f^* \in W^{r,2}(\mathcal{X})$ for some $r > 0$ and that we use the kernel associated with $\mathcal{H} = W^{m,2}(\mathcal{X})$ to run the kernel gradient flow. Then, the source condition of $f^*$ with respect to $\mathcal{H}$ is $s = r/m$. Thus, we know that the optimal convergence rate of the generalization error is $s\beta/(s\beta + 1) = 2r/(2r + d)$, which is consistent with the minimax rate of $W^{r,2}(\mathcal{X})$.

### 2.3.3   Kernel interpolation

In practice, neural networks are usually trained to a near-zero training error and have good generalization ability (Belkin et al., 2018; Zhang et al., 2021). This benign overfitting phenomenon of neural networks makes one wonder about the generalization ability of kernel interpolation

estimator $\hat{f}_{\text{inter}}$ in (8), which is also the interpolation estimator with the minimum RKHS norm:

$$\hat{f}_{\text{inter}} = \arg\min_{f \in \mathcal{H}} \|f\|_{\mathcal{H}}, \text{ s.t. } f(\boldsymbol{x}_i) = y_i, \quad i = 1, \dots, n.$$

In fixed dimensions, several works have claimed the inconsistency of $\hat{f}_{\text{inter}}$ under various settings. Rakhlin and Zhai (2019) showed that $\hat{f}_{\text{inter}}$ with the Laplace kernel is inconsistent when dimension $d$ is odd. Buchholz (2022) proved the inconsistency when $k$ is the kernel function associated with the Sobolev space $W^{m,2}(\mathcal{X}), d/2 < m < 3d/4$. Beaglehole et al. (2023a) showed the inconsistency for a class of shift-invariant periodic kernels under mild spectral assumptions. Li et al. (2023) showed that the generalization ability of kernel interpolation can be arbitrarily bad for those RKHSs that satisfy the embedding index assumption. Lai et al. (2023) showed that kernel interpolation with the neural tangent kernel of a fully connected two-layer ReLU neural network on one-dimensional data is approximately a linear interpolation. All of these results suggest that kernel interpolation can not generalize well in fixed dimensions.

Beyond inconsistency, Mallinar et al. (2022) and Cheng et al. (2024) discussed the tempered regime (the generalization error remains bounded) and the catastrophic regime (the generalization error diverges to infinity) of kernel interpolation. Haas et al. (2024) found that adding spike components to kernels could lead to consistent or even rate-optimal kernel interpolation in fixed dimensions.

## 2.4 Kernel regression in high dimensions

Since neural networks often perform well on high-dimensional data, high-dimensional kernel regression has garnered much recent interest. In this subsection, we still use the notations in Section 2.2 and summarize the results when the sample size and dimension satisfy $n \asymp d^\gamma$ for some $\gamma > 0$. Compared with the fixed-dimensional setting, as $d$ varies, the eigenvalues of $\mathcal{H}$ usually depend on $d$ in an unpleasant way. Thus, the polynomial decay rate of the eigenvalues in Assumption 1 must not hold. Most existing results considered specific kernels (e.g., inner product kernel) or special input spaces (e.g., sphere, discrete hypercube), where the eigenvalues and eigenfunctions of $\mathcal{H}$ are well understood. Nevertheless, many new phenomena have emerged in high dimensions.

### 2.4.1 Polynomial approximation barrier

We first review the "polynomial approximation barrier" of kernel regression studied by Ghorbani et al. (2021) and several subsequent publications (Donhauser et al., 2021; Ghorbani et al., 2020; Ghosh et al., 2021; Hu and Lu, 2022a; Mei et al., 2021, 2022; Misiakiewicz, 2022; Xiao et al., 2022, etc.). This line of work assumed the true function $f^*$ to be square-integrable. Specifically, Ghorbani et al. (2021) considered the inner product kernel on the sphere $\mathcal{X} = \mathbb{S}^{d-1}$ with uniform distribution, defined as

$$k(\boldsymbol{x}, \boldsymbol{x}') = h\left(\langle \boldsymbol{x}, \boldsymbol{x}' \rangle\right), \ \forall \boldsymbol{x}, \boldsymbol{x}' \in \mathbb{S}^{d-1}, \tag{12}$$

where $h(z) \in \mathcal{C}^\infty\left([-1,1]\right)$ is a fixed function independent of $d$ and

$$h(z) = \sum_{j=0}^{\infty} a_j z^j, \ a_j > 0, \ \forall j = 0, 1, 2, \dots.$$

(This definition of kernel is equivalent to assuming that Assumption 3 of Ghorbani et al. (2021) holds for all levels $\ell \in \mathbb{N}$.) The eigenfunctions of such kernels are spherical harmonic polynomials, and there exists a concise characterization of the order of eigenvalues (see, e.g., Smola et al. 2000 and Lu et al. (2023)). When $n \asymp d^\gamma, \gamma \in (0, \infty) \backslash \mathbb{N}^+$, Theorem 4 of Ghorbani et al. (2021) showed that the generalization error of the kernel ridge regression $\hat{f}_\lambda$ with $\lambda = O(n^{-1})$ satisfies (with high probability)

$$\left| \mathcal{R}(\hat{f}_\lambda) - \|\mathbf{P}_{>\ell} f^*\|_{L^2}^2 \right| \leq \varepsilon \left( \|f^*\|_{L^2}^2 + \sigma_0^2 \right),$$

where $\ell = \lfloor \gamma \rfloor$, $\mathbf{P}_{\leq \ell}$ denotes the projection operator that projects to the subspace of polynomials of degree at most $\ell$, $\mathbf{P}_{>\ell} = \mathbf{I} - \mathbf{P}_{\leq \ell}$, and $\varepsilon$ is a positive real number. The results can be viewed more intuitively as

$$\mathcal{R}(\hat{f}_\lambda) = \|\mathbf{P}_{>\ell} f^*\|_{L^2}^2 + o_{\mathbb{P}}(1).$$

They also showed that $\|\mathbf{P}_{>\ell} f^*\|_{L^2}^2 + o_{\mathbb{P}}(1)$ is the best generalization error achievable by a kernel regression in the form of $\hat{f}(\boldsymbol{x}) = \sum_{i=1}^n a_i k(\boldsymbol{x}, \boldsymbol{x_i}), a_i \in \mathbb{R}$. This polynomial approximation barrier was later used to quantify advantages of feature learning in some literature (see Section 3.3). In the case of $\gamma \in \mathbb{N}^+$, which was not covered by Ghorbani et al. (2021), subsequent works Xiao et al. (2022), Hu and Lu (2022a), and Misiakiewicz (2022) derived the precise asymptotic formulas for the bias and variance terms of the kernel ridge regression and showed that the generalization error achieved the peak when $n \approx d^\gamma / \gamma!, \gamma \in \mathbb{N}^+$. Mei et al. (2022) studied a similar approximation barrier for kernel regression for kernels whose eigenspaces have hypercontractivity and satisfy certain spectral conditions.

The key idea of this line of work is to decompose the empirical kernel matrix $\mathbb{K}(\boldsymbol{X}, \boldsymbol{X})$ into low-frequency and high-frequency parts, then to take advantage of the nice properties of the eigenfunctions (which are spherical harmonic polynomials for the inner kernel of the product on the sphere) when $d$ is large enough.

### 2.4.2 Generalization behaviors under the source condition

Another line of work considered the source condition (Assumption 2) in the high-dimensional setting (Liu et al., 2021; Lu et al., 2023; Zhang et al., 2024b). Compared with the work mentioned in Section 2.4.1, which only assumes $f^*$ to be square-integrable, $[\mathcal{H}]^s$ is a smaller function space than $L^2(\mathcal{X}, \mu)$ when $s > 0$, and $L^2(\mathcal{X}, \mu)$ can be viewed as a limiting case of $s \to 0$.

Next, we review the results in Zhang et al. (2024b), which also studied the inner product kernel on the sphere (12), provided minimax optimal rates, and found some new phenomena for the kernel ridge regression. Note that two parameters, $s$ and $\gamma$, determine the generalization ability. Zhang et al. (2024b) showed that when $s \in (0, 1)$, $s \in [1, 2]$ and $s > 2$, the generalization error behaved differently along $\gamma$ (Zhang et al. 2024b, Theorems 2 and 3). Specifically, when $0 < s < 1$, the generalization error under the best choice of the regularization parameter (denoted as $R^*$) has two periods:

(i) if $\gamma \in (p + ps, p + ps + s], p \in \mathbb{N}$,

$$R^* = \Theta_{\mathbb{P}} \left( d^{-\gamma + p} \right) = \Theta_{\mathbb{P}} \left( n^{-1 + \frac{p}{\gamma}} \right);$$

(ii) if $\gamma \in (p + ps + s, (p+1) + (p+1)s]$, $p \in \mathbb{N}$,

$$R^* = \Theta_{\mathbb{P}}\left(d^{-(p+1)s}\right) = \Theta_{\mathbb{P}}\left(n^{-\frac{(p+1)s}{\gamma}}\right).$$

When $1 \leq s \leq 2$, $R^*$ has three periods:

(i) if $\gamma \in (p + ps, \; p + ps + 1]$, $p \in \mathbb{N}$,

$$R^* = \Theta_{\mathbb{P}}\left(d^{-\gamma+p}\right) = \Theta_{\mathbb{P}}\left(n^{-1+\frac{p}{\gamma}}\right);$$

(ii) if $\gamma \in (p + ps + 1, \; p + ps + 2s - 1]$, $p \in \mathbb{N}$,

$$R^* = \Theta_{\mathbb{P}}\left(d^{-\frac{\gamma-p+ps+1}{2}}\right) = \Theta_{\mathbb{P}}\left(n^{-\frac{\gamma-p+ps+1}{2\gamma}}\right).$$

(iii) if $\gamma \in (p + ps + 2s - 1, (p+1) + (p+1)s]$, $p \in \mathbb{N}$,

$$R^* = \Theta_{\mathbb{P}}\left(d^{-(p+1)s}\right) = \Theta_{\mathbb{P}}\left(n^{-\frac{(p+1)s}{\gamma}}\right).$$

When $s > 2$, $R^*$ is exactly the same as $s = 2$. Furthermore, Zhang et al. (2024b), Theorem 5, provided the corresponding minimax lower rate of $[\mathcal{H}]^s, s > 0$, which is omitted here. For instance, Figure 2 shows the best convergence rates of KRR and the corresponding minimax lower rates (with respect to $d$) for $s = 1.5$ and $\gamma > 0$. More visualizations of these convergence rates can be found in Zhang et al. (2024b), Figures 2 and 3.

Several interesting phenomena arise from these results.

- Periodic plateau behavior: when $\gamma$ varies within a certain period, the rate with respect to dimension $d$ does not change with $\gamma$.

- Multiple descent behavior: the rates with respect to sample size $n$ achieve peaks and isolated valleys at certain values of $\gamma$ (Zhang et al. 2024b, Figure 3).

- Minimax optimality and new saturation effect: when $0 < s \leq 1$, the best convergence rate of KRR matches the minimax lower rate for all $\gamma > 0$. When $s > 1$, KRR cannot achieve the minimax lower rate for certain ranges of $\gamma$, which is called the new saturation effect of KRR.

The periodic plateau and multiple-descent behaviors were first reported in Lu et al. (2023) for kernel gradient flow with $s = 1$. They also included an example of the neural tangent kernel of the two-layer fully connected ReLU neural network.

### 2.4.3 Kernel interpolation

We have shown in Section 2.3.3 that kernel interpolation $\hat{f}_{\text{inter}}$ cannot generalize in fixed dimensions. However, kernel interpolation can generalize surprisingly well in high dimensions, as first demonstrated theoretically by Liang and Rakhlin (2020). Liang and Rakhlin (2020) studied the inner product kernel and the setting $n \asymp d$. Using the linear approximation of high-dimensional kernel matrices from Karoui (2010), Liang and Rakhlin (2020) first proved

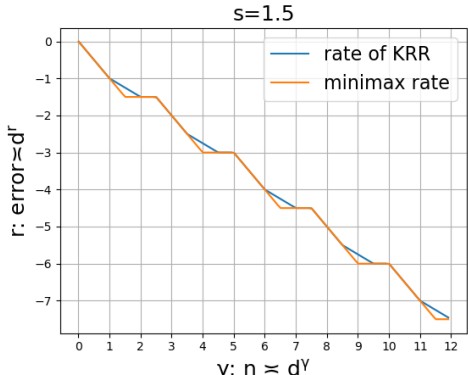

Figure 2: Best convergence rates of KRR and corresponding minimax lower rate (w.r.t. $d$) for $s = 1.5$ and $\gamma > 0$.

the upper bounds of the bias and variance terms of kernel interpolation, and showed that the generalization error will converge to zero when the data exhibit a low-dimensional structure.

Liang et al. (2020) also considered the inner product kernel and assumed that the coordinates of the $d$-dimensional input are independent and identically distributed. Considering the high-dimensional setting $n \asymp d^\gamma, \gamma > 0$, they proved an upper bound of the variance term. Furthermore, assuming $f^*(\boldsymbol{x}) = \langle k(\boldsymbol{x}, \cdot), \rho(\cdot) \rangle_{L^2}$, with $\|\rho\|_{L^4}^4 \leq C$ for some constant $C > 0$, they demonstrated that the bias term is infinitesimal of a higher order compared with the variance term. They obtained the following upper bound for the generalization error with a concrete convergence rate (denote $l = \lfloor \gamma \rfloor$):

$$\mathcal{R}(\hat{f}_{\text{inter}}) = O_{\mathbb{P}} \left( d^{l-\gamma} + d^{\gamma-l-1} \right).$$

This upper bound exhibits multiple-descent behavior, i.e., the convergence rate is non-monotone as $\gamma$ increases. They also provided visualization and empirical evidence for the multiple-descent behavior (see Liang et al. 2020, Figures 1 and 2).

To our knowledge, Aerni et al. (2022) is the first to provide lower bounds of the bias and variance terms for kernel interpolation in high dimensions. They considered the convolutional kernel on the discrete hypercube $\{-1, 1\}^d$ and a special form of the true function $f^*(x) = x_1 x_2 \cdots x_{L^*}$, where $L^*$ is formulated in Aerni et al. (2022), Theorem 1. Using a similar decomposition of the empirical kernel matrix $\mathbb{K}(\boldsymbol{X}, \boldsymbol{X})$ (as mentioned at the end of Section 2.4.1) as the line of work Ghorbani et al. (2021), this $f^*$ will fall into the eigenspace corresponding to the low-frequency part. They demonstrated that kernel interpolation has generalization ability in this setting and discovered the multiple-descent behavior of kernel interpolation.

Furthermore, Zhang et al. (2024c) showed that whether kernel interpolation is a good choice (in the sense of minimax optimality) depends on the relative smoothness of the true function. Specifically, they considered the inner product kernel on the sphere, $n \asymp d^\gamma, \gamma > 0$, and assumed the source condition of $f^*$ to be exactly $s \geq 0$. For all values of $s \geq 0$ and $\gamma > 0$, they fully characterized the exact orders of the bias and variance terms, leading to an exact order of the generalization error: (denote $l = \lfloor \gamma \rfloor$)

$$\mathcal{R}(\hat{f}_{\text{inter}}) = \Theta_{\mathbb{P}} \left( d^{l-\gamma} + d^{\gamma-l-1} + d^{-(l+1)s} \right).$$

Comparing this rate with the minimax lower rate in $[\mathcal{H}]^s$ in Zhang et al. (2024b), they further showed that for different values of $(s, \gamma)$, kernel interpolation can be minimax optimal, consistent but sub-optimal, or inconsistent (see Figure 3, borrowed from Zhang et al. 2024c). Specifically, for any fixed $\gamma \in (0, \infty) \backslash \mathbb{N}^+$, there exists a threshold $\Gamma(\gamma)$ (see Zhang et al. 2024c, Eq.(11)) such that when $s > \Gamma(\gamma)$, kernel interpolation is sub-optimal; when $0 < s \leq \Gamma(\gamma)$, kernel interpolation is minimax optimal; and when $s = 0$ or $\gamma \in \mathbb{N}^+$, kernel interpolation is inconsistent. The existence of a threshold $\Gamma(\gamma)$ provides a comprehensive answer to the question "when does the benign overfitting phenomenon occur" in kernel regression and shows how it depends on the relative smoothness of the true function and the high-dimensional scaling.

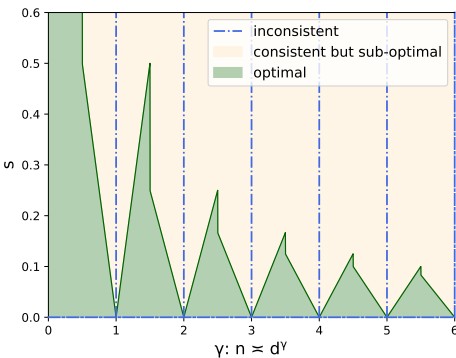

Figure 3: Phase diagram about the consistency and the optimality of kernel interpolation.

### 2.4.4 Other topics

Recall that most of the results discussed above focused on specific kernels and input spaces. Barzilai and Shamir (2023) and Misiakiewicz and Saeed (2024) aimed to provide a unified theory of the generalization error of kernel regression under mild assumptions. In addition to general kernel ridge regression bounds, Gavrilopoulos et al. (2024) analyze a conjugate kernel obtained after training its feature map on the data; we compare this mechanism with the adaptive feature model in Section 4.1. It is also worth mentioning the study of benign overfitting in high-dimensional linear regression with an arbitrary structure of the covariance of the input data (Bartlett et al., 2020; Hastie et al., 2022; Kobak et al., 2020; Tsigler and Bartlett, 2023, etc.). It can be seen that high-dimensional linear regression and kernel regression share many common characteristics.

## 3 From fixed features to feature learning

### 3.1 Limitations of fixed features

Despite significant successes of the neural tangent kernel (NTK) theory, an increasing number of researchers argue that it does not explain the outstanding performance of real-world neural networks. Notable works highlighting these limitations include, but are not limited to, Wei et al. (2019) and Ghorbani et al. (2019, 2020) from the theoretical perspective; Chizat et al. (2019) and

Arora et al. (2019b) from the empirical perspective. The most widely discussed shortcoming of the NTK theory is its lack of feature learning capability, since it equates the training dynamics of neural networks entirely with kernel regression using the specific neural tangent kernel. In the following, we analyze the limitations of the NTK theory from multiple perspectives.

For a kernel function $k$, Mercer's decomposition (5) can be written as

$$k(\boldsymbol{x}, \boldsymbol{x}') = \Psi(\boldsymbol{x})^\top \boldsymbol{\Lambda} \Psi(\boldsymbol{x}'),$$

where we denote $\Psi(\boldsymbol{x}) = (\psi_1(\boldsymbol{x}), \psi_2(\boldsymbol{x}), \ldots)^\top$ and $\boldsymbol{\Lambda} = \mathrm{Diag}\{\lambda_1, \lambda_2, \ldots\}$. Define the feature map:

$$\boldsymbol{x} \to \Phi(\boldsymbol{x}) = \boldsymbol{\Lambda}^{\frac{1}{2}} \Psi(\boldsymbol{x}). \tag{13}$$

The kernel function is then the inner product of the two features at $\boldsymbol{x}$ and $\boldsymbol{x}'$, i.e.,

$$k(\boldsymbol{x}, \boldsymbol{x}') = \Phi(\boldsymbol{x})^\top \Phi(\boldsymbol{x}'). \tag{14}$$

It is well known that kernel regression is equivalent to a linear regression in the feature space. Specifically, (i) the kernel ridge regression estimator in Definition 2.2.1 is the same as the ridge regression estimator in the feature space; (ii) the kernel gradient flow estimator in Definition 2.2.2 can be expressed as $\hat{f}_t(\boldsymbol{x}) = \Phi(\boldsymbol{x})^\top \boldsymbol{b}_t$, where $\boldsymbol{b}_t$ is obtained by a gradient flow to minimize square loss $\mathcal{L}(\boldsymbol{b}) = \|\boldsymbol{Y} - \Phi(\boldsymbol{X})^\top \boldsymbol{b}\|_2^2 / n$. Once the architecture of a neural network is determined, the feature corresponding to the NTK is fixed and independent of the data. Therefore, kernel regression with NTK disregards neural networks' ability to learn features from the data.

In the classical nonparametric regimes reviewed in Section 2.3.2, an appropriately regularized fixed kernel method can attain the minimax rate when the kernel and source condition are matched. However, the minimax rate is a worst-case benchmark over a function class and need not coincide with the best achievable rate for a specific $f^*$. When a pre-specified feature map is poorly aligned with $f^*$, as in the high-dimensional polynomial approximation barrier of Section 2.4.1, feature learning can offer an advantage if training changes the representation to improve that alignment.

Additionally, the results in Sections 2.3.3 and 2.4.3 show that kernel interpolation cannot generalize in fixed dimensions and does not outperform regularized kernel regression (unless $f^*$ is less smooth) in high dimensions. This also suggests that kernel regression may not be an appropriate method to interpret "the benign overfitting phenomenon", which is widely observed for neural networks (Zhang et al., 2021).

In the NTK theory, a special symmetric initialization of the parameters is widely used (Chizat et al., 2019; Hu et al., 2019; Lai et al., 2023), ensuring that the output of a neural network at initialization is zero, e.g., $f_{\boldsymbol{\theta}(0)}^m(\boldsymbol{x}) \equiv 0$ in (4). Recall that the explicit expression of the KGF estimator (7) also relies on the zero initialization $\hat{f}_0(\boldsymbol{x}) \equiv 0$. Recent work (Chen et al., 2024) showed that, in the NTK regime, nonzero initialization of a neural network would introduce a terrible bias. Specifically, they showed that a sufficiently wide and fully connected neural network with each parameter initialized as an independent standard normal variable achieves an optimal convergence rate of generalization error $n^{-\frac{3}{d+3}}$, no matter how smooth $f^*$ is (in the sense of the source condition $f^* \in [\mathcal{H}]^s$). This convergence rate falls into the "curse of dimensionality": As the dimensionality $d$ increases, the sample size $n$ required for good generalization ability

increases exponentially. Thus, they suggested that there is a divergence between the NTK theory and real-world neural networks.

## 3.2 Implications of fixed features

Although the theory of fixed features has limitations in explaining real-world neural networks, it offers insights into what constitutes a good feature representation. In this subsection, as a warm-up of the adaptive feature approach, we review two examples to illustrate a key observation:

*The alignment between the true function and the chosen feature matters.*

More precisely, for a given kernel feature map $\Phi(\boldsymbol{x})$ (defined in (13)) and the corresponding kernel $k$, we say that the feature map is more aligned with the true function $f^*$ when a larger share of the squared projection energy $\langle f^*, \psi_j \rangle_{L^2}^2$ lies in eigenspaces associated with larger eigenvalues $\lambda_j$. This is a property of a feature representation fixed before training. For the high-dimensional inference problems of Ben Arous et al. (2021), the information exponent governs the sample complexity of an initial phase in which online SGD obtains nontrivial correlation with the target. Abbe et al. (2023, 2024) study related two-layer SGD settings in which learnability depends on hierarchical structure in the target. By contrast, Lecué et al. (2025) formalize alignment and feature learning through a feature space decomposition: the spectral methods they study can exploit alignment in a fixed representation but do not learn a feature map that depends on the target. Thus, alignment helps a fixed kernel only when the relevant directions are already available in its feature space, whereas feature learning concerns how training acquires useful correlation with the target.

### 3.2.1 An alignment example with two kernels

Consider the following example of a one-dimensional regression problem: suppose that $\mathcal{X} = [0, 1]$ with uniform distribution. Consider a true function $f^*(x) = \sin(2\pi x)$, and apply kernel regression with the following two kernels:

$$k_1(x, x') = \mathbf{1}_{\{x \geq x'\}}(1 - x)x' + \mathbf{1}_{\{x < x'\}}(1 - x')x,$$

and

$$k_2(x, x') = \min\{x, x'\}.$$

The results of Wainwright (2019) and Li et al. (2022) give rise to explicit formulas for the eigenvalues and eigenfunctions of $k_1, k_2$, i.e., for $j = 1, 2, \ldots$

$$\lambda_{1,j} = \frac{1}{\pi^2 j^2}, \quad \psi_{1,j} = \sqrt{2} \sin(j\pi x),$$

and

$$\lambda_{2,j} = \frac{4}{\pi^2 (2j - 1)^2}, \quad \psi_{2,j}(x) = \sqrt{2} \sin\left(\frac{2j - 1}{2}\pi x\right).$$

Therefore, $k_1, k_2$ have the same decay rates of eigenvalues, $\beta_1 = \beta_2 = 2$. Recalling the definition of an interpolation space $[\mathcal{H}]^s$ in (9), since the projection $\langle f^*, \psi_{1,j} \rangle$ is nonzero only when $j = 2$, $f^*$ has the source condition $s_1 = \infty$ with respect to $k_1$. Since the projections $\langle f^*, \psi_{2,j} \rangle \asymp j^{-2}$,

$f^*$ has the source condition $s_2 = 1.5$ with respect to $k_2$. Using the results (11) in Section 2.3.2, we obtain the optimal convergence rates as $r_1 = n^{-1}, r_2 = n^{-0.75}$, respectively, when using $k_1, k_2$ to estimate $f^*$. This implies that the feature $\Phi_1$ (or kernel $k_1$), which aligns more with $f^*$, is a better choice than $\Phi_2$ (or $k_2$).

### 3.2.2 Alignment under the neural tangent kernel

The second example, from Arora et al. (2019a), considered the two-layer fully connected ReLU neural network. Denote the noiseless labels as $\boldsymbol{Y} = (y_1, \ldots, y_n)^\top$, and let the empirical NTK matrix $\mathbb{K} \in \mathbb{R}^{n \times n}$ have entries $\mathbb{K}_{ij} = k(\boldsymbol{x}_i, \boldsymbol{x}_j)$, $i, j = 1, \ldots, n$, where $k$ is the NTK of the fully connected two-layer ReLU neural network. Then under certain assumptions (the width of the hidden layer grows sufficiently fast, and the neural network is trained for a sufficiently long time), the generalization error of the neural network estimator $\hat{f}_{\mathrm{NN}}$ has an upper bound:

$$\mathcal{R}(\hat{f}_{\mathrm{NN}}) = O_{\mathbb{P}}\left(\sqrt{\frac{\boldsymbol{Y}^\top \mathbb{K}^{-1} \boldsymbol{Y}}{n}}\right). \tag{15}$$

Let the eigenvalues and eigenvectors of the matrix $\mathbb{K}$ be $\hat{\lambda}_j, \hat{\boldsymbol{v}}_j$, i.e., $\mathbb{K} = \sum_{j=1}^n \hat{\lambda}_j \hat{\boldsymbol{v}}_j \hat{\boldsymbol{v}}_j^\top$, we have

$$\boldsymbol{Y}^\top \mathbb{K}^{-1} \boldsymbol{Y} = \sum_{j=1}^n \frac{1}{\hat{\lambda}_j} \left(\hat{\boldsymbol{v}}_j^\top \boldsymbol{Y}\right)^2. \tag{16}$$

$\hat{\lambda}_j, \hat{\boldsymbol{v}}_j$ are empirical versions of the eigenvalues and eigenfunctions of $k$. Also note that they considered the noiseless case, so $\boldsymbol{Y}$ is an empirical version of the true function $f^*$. Therefore, (15) and (16) imply that projections $\langle f^*, \psi_j \rangle_{L^2}$ corresponding to small eigenvalues $\lambda_j$ must be small for a good generalization ability.

## 3.3 Recent advances in feature learning

The aim of the adaptive feature approach in the next section is to theoretically analyze the feature learning process of neural networks and its advantages. Before that, we summarize some of the recent advances in feature learning.

There is a line of work empirically studying the evolution of the alignment between labels and features during the training of neural networks (Atanasov et al., 2021; Baratin et al., 2021; Fort et al., 2020; Kopitkov and Indelman, 2020; Maennel et al., 2020; Ortiz-Jiménez et al., 2021; Oymak et al., 2019; Shan and Bordelon, 2021, etc.). Some of these papers considered an index called "(centered) kernel alignment" (Cortes et al., 2012; Kornblith et al., 2019) under various settings and observed that the alignment increases as training progresses. Moreover, even for neural networks with infinite width, the parameterization, initialization scale, and learning rate scaling jointly determine whether the training dynamics remain in a kernel regime or exhibit feature learning. In particular, the mean field limits (Bordelon and Pehlevan, 2022; Chizat et al., 2019; Geiger et al., 2020; Woodworth et al., 2020; Yang and Hu, 2020) show that the alignment can still increase during training. In these scenarios, feature learning refers to the phenomenon that the parameters of neural networks evolve non-trivially and cannot be considered approximately unchanged, which is in contrast to the frozen feature in the neural tangent kernel (NTK) regime.

Theoretically, characterizing what feature can be learned from the data and analyzing its impact on the method's generalization ability is a challenge. One line of work studies recursive feature machines to provide an explicit adaptive kernel construction: in the fully connected and convolutional settings studied by Beaglehole et al. (2023b); Radhakrishnan et al. (2022), the feature map is updated from the data rather than held fixed as in an NTK approximation. A notable line of work dealt with the problem by considering the one-step gradient descent setting (Ba et al., 2022; Cui et al., 2024; Damian et al., 2022; Dandi et al., 2023; Moniri et al., 2023). These works originated from the random feature model (Gerace et al., 2020; Hu and Lu, 2022b; Mei and Montanari, 2022; Rahimi and Recht, 2007), etc., and aimed to prove that feature learning brought by one-step gradient descent can lead to advantages over random features and kernel regression estimators. Next, we briefly introduce the results in Ba et al. (2022). They considered the following two-layer fully connected neural network

$$f_{\mathrm{NN}}(\boldsymbol{x}) = \frac{1}{\sqrt{m}} \sum_{r=1}^{m} a_r \sigma\left(\boldsymbol{w}_r^\top \boldsymbol{x}\right) = \frac{1}{\sqrt{m}} \boldsymbol{a}^\top \sigma\left(\boldsymbol{W}^\top \boldsymbol{x}\right), \tag{17}$$

and the proportional asymptotic limits, i.e., the width $m$, sample size $n$ and sample dimension $d$ all tend to infinity and satisfy

$$n/d \to \gamma_1, m/d \to \gamma_2, \quad \gamma_1, \gamma_2 \in (0, \infty).$$

Denoting $\boldsymbol{W}_0$ as the weight matrix at initialization, they first updated $\boldsymbol{W}_0$ for one step:

$$\boldsymbol{W}_1 = \boldsymbol{W}_0 + \eta\sqrt{m} \cdot \boldsymbol{G}_0,$$

where $\boldsymbol{G}_0$ is the gradient of the square loss with respect to $\boldsymbol{W}_0$, and $\eta$ is the learning rate. Then, a ridge regression was conducted in the new feature space $\boldsymbol{x} \to \sigma\left(\boldsymbol{W}_1^\top \boldsymbol{x}\right)$. They also assumed that the true function was a single index function $f^*(\boldsymbol{x}) = \sigma^*(\langle \boldsymbol{x}, \boldsymbol{\beta}_* \rangle)$, where $\boldsymbol{x} \sim \mathcal{N}(\boldsymbol{0}, \mathbf{I}_d)$ and $\|\boldsymbol{\beta}_*\|_2 = 1$. Denote $\mathcal{R}_0(\lambda)$ and $\mathcal{R}_1(\lambda)$ as the generalization errors of the ridge regression estimators with the regularization parameter $\lambda > 0$ using the features $\sigma\left(\boldsymbol{W}_0^\top \boldsymbol{x}\right)$ and $\sigma\left(\boldsymbol{W}_1^\top \boldsymbol{x}\right)$, respectively. Under some detailed assumptions, by choosing the learning rate $\eta = \Theta(1)$, they proved that (Ba et al. 2022, Theorem 5):

$$\mathcal{R}_0(\lambda) - \mathcal{R}_1(\lambda) \xrightarrow{\mathbb{P}} \delta \geq 0,$$

where $\delta$ is constant with explicit expression in their paper. For this constant learning rate $\eta = \Theta(1)$, one-step gradient descent has already shown an improvement over the initial random feature. Furthermore, by choosing a larger learning rate $\eta = \Theta(\sqrt{m})$, they proved an upper bound of $\mathcal{R}_1(\lambda)$ (Ba et al. 2022, Theorem 7), which outperforms the kernel lower bound $\|\mathbf{P}_{>1} f^*\|_{L^2}^2$ (Hu and Lu 2022b; Montanari and Zhong 2022, the same as the polynomial approximation barrier in Section 2.4.1) for some examples.

Something more interesting was also studied about the first step gradient $\boldsymbol{G}_0$ and the new feature $\boldsymbol{W}_1$. They showed that $\boldsymbol{G}_0$ is close to a rank-1 matrix $\boldsymbol{X}^\top \boldsymbol{Y} \boldsymbol{a}^\top / n\sqrt{m}$ (omitting the constant), where $\boldsymbol{a}$ is the weights of the randomly initialized output layer, and $\boldsymbol{X}^\top \boldsymbol{Y}$ is roughly $\boldsymbol{\beta}_*$ in their true function. Furthermore, they showed that the first singular vector $\boldsymbol{u}_1$ of $\boldsymbol{W}_1$ (corresponding to the leading singular value) satisfies the following:

$$|\langle \boldsymbol{u}_1, \boldsymbol{\beta}_* \rangle|^2 \xrightarrow{\mathbb{P}} C_0,$$

where the expression $C_0 \in (0,1)$ is provided in their paper. It can be seen from their expression that $C_0 \to 1$ when $\gamma_1 \to \infty$, and $C_0$ increases as the learning rate increases. Despite many specific assumptions, they theoretically proved that the alignment (similar to the alignment discussed at the beginning of Section 3.2) between the feature and labels emerges from the training of neural networks, and showed the advantage of feature learning.

There is a substantial body of literature studying feature learning, which is too extensive to list comprehensively here. For instance, Hanin and Nica (2019), Dyer and Gur-Ari (2019), Huang and Yau (2020), Yaida (2020), Naveh and Ringel (2021), and Bordelon and Pehlevan (2024) studied the finite-width corrections which enhanced feature evolving. Toward analyzing feature learning with fewer assumptions on the true function, Radhakrishnan et al. (2022), Beaglehole et al. (2023b), Beaglehole et al. (2024), and Radhakrishnan et al. (2024) proposed a general structure of the weights of neural networks during training, which was called the neural feature ansatz.

# 4 Adaptive feature model and over-parameterized Gaussian sequence

## 4.1 Neural networks as adaptive feature model

As discussed in Sections 3.1 and 3.2, kernel regression is equivalent to linear regression in its associated feature space, and the alignment between the feature and the target function affects generalization. Neural networks can likewise be viewed as linear regressions on features that are learned during training. Throughout this section, we consider one-dimensional outputs. Let $\boldsymbol{\beta} \in \mathbb{R}^m$ denote the last-layer weights and $\Phi_{\boldsymbol{\eta}}(\boldsymbol{x}) \in \mathbb{R}^m$ the feature before the last layer. The network output then has the form

$$f_{\boldsymbol{\theta}}(\boldsymbol{x}) = \Phi_{\boldsymbol{\eta}}(\boldsymbol{x})^\top \boldsymbol{\beta}, \tag{18}$$

where $\boldsymbol{\theta} = \text{vec}(\{\boldsymbol{\beta}, \boldsymbol{\eta}\})$ collects the learnable parameters of the neural network. For example, in (17), $\boldsymbol{\beta} = \boldsymbol{a}$, $\Phi_{\boldsymbol{\eta}}(\boldsymbol{x}) = \sigma(\boldsymbol{W}^\top \boldsymbol{x})/\sqrt{m}$, and $\boldsymbol{\eta} = \boldsymbol{W}$.

Although the parameterization of $\Phi_{\boldsymbol{\eta}}(\boldsymbol{x})$ is architecture dependent and can be complicated, the estimator in (18) admits an abstract representation as a more general *adaptive feature model*.

**Definition 4.1.1** (Adaptive feature model). *Consider the nonparametric regression problem* (1). *For a learnable feature $\Phi_{\boldsymbol{\eta}}(\boldsymbol{x}) \in \mathbb{R}^m$ with parameters $\boldsymbol{\eta}$, define the adaptive feature estimator by*

$$f_t(\boldsymbol{x}) = \Phi_{\boldsymbol{\eta}_t}(\boldsymbol{x})^\top \boldsymbol{\beta}_t,$$

*where $\boldsymbol{\eta}_t$ and $\boldsymbol{\beta}_t$ are obtained at time $t$ by gradient flow minimizing the loss*

$$\mathcal{L}(\boldsymbol{\eta}, \boldsymbol{\beta}) = \left\| \boldsymbol{Y} - \Phi_{\boldsymbol{\eta}}(\boldsymbol{X})^\top \boldsymbol{\beta} \right\|_2^2. \tag{19}$$

The feature dimension may be finite or infinite. By analogy with (14), the associated time-varying kernel is

$$k_{\boldsymbol{\eta}}(\boldsymbol{x}, \boldsymbol{x}') := \Phi_{\boldsymbol{\eta}}(\boldsymbol{x})^\top \Phi_{\boldsymbol{\eta}}(\boldsymbol{x}').$$

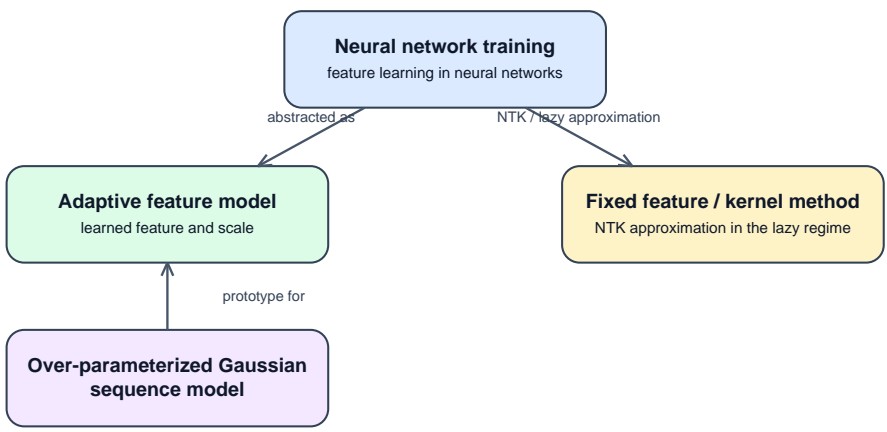

Figure 4: A conceptual map linking neural network training, the adaptive feature model, the NTK/fixed-feature approximation, and the over-parameterized Gaussian sequence model.

The adaptive feature model in Definition 4.1.1 gives an abstract description of feature learning in neural networks, but the training dynamics in (19) remain difficult to analyze. The main difficulties are:

(i) the parameterization of $\Phi_{\boldsymbol{\eta}}(\boldsymbol{x})$ with respect to $\boldsymbol{\eta}$ is usually complicated;

(ii) the evolution of $\Phi_{\boldsymbol{\eta}}(\boldsymbol{x})$ and $\boldsymbol{\beta}$ is coupled during training.

The neural tangent kernel (NTK) theory addresses the first difficulty by considering a sufficiently wide regime in which the feature remains approximately unchanged during training, i.e., $\Phi_{\boldsymbol{\eta}_t}(\boldsymbol{x}) \approx \Phi_{\boldsymbol{\eta}_0}(\boldsymbol{x})$. The works Ba et al. (2022), Damian et al. (2022), Dandi et al. (2023), Moniri et al. (2023), and Cui et al. (2024), discussed in Section 3.3, simplify the second difficulty by analyzing one step of gradient descent. Lecué et al. (2025) give formal definitions of alignment and feature learning, whereas Gavrilopoulos et al. (2024) analyze a specified mechanism with two stages through a learned conjugate kernel and its excess risk. The model in Definition 4.1.1 instead serves as an abstraction of coupled feature and coefficient dynamics, and the over-parameterized Gaussian sequence model below is a simpler prototype that isolates rotation and rescaling. The following subsections develop an alternative approach to studying the dynamics in (19) while retaining, as far as possible, the feature learning characteristics of neural networks. Figure 4 gives a conceptual map of the relations among adaptive feature models, neural network training, the NTK approximation, and the Gaussian sequence model prototype.

We next present experiments showing that the feature learned by a neural network ($\Phi_{\boldsymbol{\eta}}(\boldsymbol{x})$ in (18)) gradually aligns with the data during training. Recall that Section 3.2 characterizes a better feature vector $\Phi(\boldsymbol{x}) = \boldsymbol{\Lambda}^{\frac{1}{2}}\Psi(\boldsymbol{x})$ by the concentration of the projections $\langle f^*, \psi_j \rangle_{L^2}$ on eigenspaces with larger eigenvalues $\lambda_j$. Let $\hat{\boldsymbol{v}}_j \in \mathbb{R}^n$ denote the $j$-th singular vector of the neural network feature $\Phi_{\boldsymbol{\eta}}(\boldsymbol{X})$, with singular values arranged in descending order. Then $\sqrt{n}\hat{\boldsymbol{v}}_j$ serves as an approximation to the $j$-th eigenfunction of $\Phi_{\boldsymbol{\eta}}(\boldsymbol{x})$. With $\boldsymbol{Y} \in \mathbb{R}^n$ denoting the sample labels, we use $f_j := \frac{1}{\sqrt{n}}\hat{\boldsymbol{v}}_j^\top \boldsymbol{Y}$ to approximate $\langle f^*, \psi_j \rangle_{L^2}$. Figure 5 plots the percentage of the

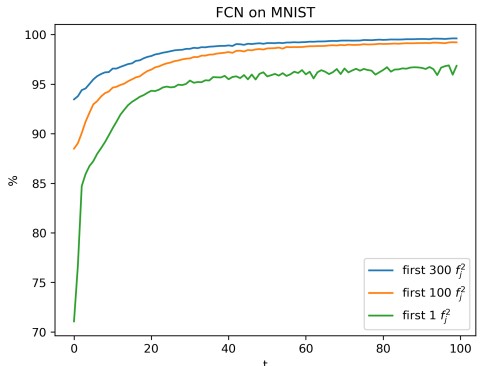 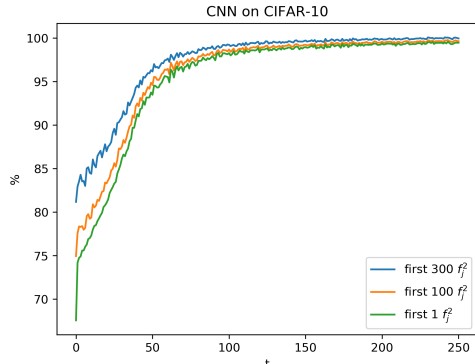

Figure 5: $x$-axis: number of training iterations; $y$-axis: the percentage of the first $p$ projections: $\sum_{j=1}^{p} f_j^2 / \sum_{j=1}^{m} f_j^2$, p = 1, 100, 300, m = 500. The projections $f_j$ concentrate on top eigenspaces as the training proceeds.

first $p$ projections, $\sum_{j=1}^{p} f_j^2 / \sum_{j=1}^{m} f_j^2$, at each iteration $t$, for $p = 1$, 100, and 300. Here $m$ is the dimension of $\Phi_{\boldsymbol{\eta}}(\boldsymbol{x})$, and we set $m = 500$ in the experiments. The percentage increases over training, indicating improved alignment between the neural network feature and the data. We use a fully connected two-layer neural network (FCN) on MNIST and a three-layer convolutional neural network (CNN) on CIFAR-10. Further experimental details appear in Appendix.

## 4.2 Relation between kernel regression and the Gaussian sequence model

The preceding subsection introduced the adaptive feature model and the difficulty of studying its training dynamics. Section 4.3 introduces the *over-parameterized Gaussian sequence model* as a prototype for the adaptive feature model. We view the adaptive feature model as an adaptive version of kernel regression with fixed features, and the "over-parameterized Gaussian sequence" model as an adaptive version of the *Gaussian sequence model* (Johnstone, 2017).

Let $k(\boldsymbol{x}, \boldsymbol{x}') = \Psi(\boldsymbol{x})^\top \boldsymbol{\Lambda} \Psi(\boldsymbol{x}')$, where $\Psi(\boldsymbol{x}) = (\psi_1(\boldsymbol{x}), \psi_2(\boldsymbol{x}), \ldots)^\top$ is an orthonormal basis of eigenfunctions in $L^2(\mathcal{X}, \mu)$ and $\boldsymbol{\Lambda} = \mathrm{Diag}\{\lambda_1, \lambda_2, \ldots\}$ contains the associated eigenvalues in decreasing order. Write the target function as $f^*(\boldsymbol{x}) = \sum_{j=1}^{\infty} \theta_j^* \psi_j(\boldsymbol{x})$. Rescale the coefficients by setting $\beta_j^* = \lambda_j^{-\frac{1}{2}} \theta_j^*$. Thus, $f^*(\boldsymbol{x}) = \sum_{j=1}^{\infty} \lambda_j^{\frac{1}{2}} \beta_j^* \psi_j(\boldsymbol{x})$. For each $j = 1, 2, \ldots$, let $\beta_{j,t}$ estimate $\beta_j^*$ and let $\theta_{j,t} = \lambda_j^{\frac{1}{2}} \beta_{j,t}$ be the corresponding estimate of $\theta_j^*$.

Kernel gradient flow with kernel $k$, equivalently linear regression in the feature space, evolves $\boldsymbol{\beta}_t = (\beta_{1,t}, \beta_{2,t}, \ldots)^\top$ to minimize the square loss:

$$
\mathcal{L}(\boldsymbol{\beta}) = \frac{1}{2n} \left\| \boldsymbol{Y} - \Psi(\boldsymbol{X})^\top \boldsymbol{\Lambda}^{\frac{1}{2}} \boldsymbol{\beta} \right\|_2^2,
$$
$$
\text{i.e., } \frac{\mathrm{d}}{\mathrm{d}t} \boldsymbol{\beta}_t = -\nabla_{\boldsymbol{\beta}} \mathcal{L}(\boldsymbol{\beta}_t),
\tag{20}
$$

where $\Psi(\boldsymbol{X}) \in \mathbb{R}^{\infty \times n}$, and the resulting estimator of $f^*$ is $\hat{f}_t = \Psi(\boldsymbol{X})^\top \boldsymbol{\Lambda}^{\frac{1}{2}} \boldsymbol{\beta}_t$.

The corresponding Gaussian sequence model observes a noisy sequence $\{z_j\}_{j=1}^{\infty}$ generated by:

$$
z_j = \theta_j^* + \xi_j, \ j \in \mathbb{N}^+; \ \xi_j \overset{\text{i.i.d.}}{\sim} \mathcal{N}(0, n^{-1}),
\tag{21}
$$

where $\{\theta_j^*\}_{j=1}^\infty$ are the projection coefficients $\langle f^*, \psi_j \rangle_{L^2}$ of $f^*$. Consider the gradient flow of $\boldsymbol{\beta}_t = (\beta_{1,t}, \beta_{2,t}, \ldots)^\top$ minimizing the square loss:

$$\mathcal{L}(\boldsymbol{\beta}) = \frac{1}{2} \sum_{j=1}^\infty \left( z_j - \lambda_j^{\frac{1}{2}} \beta_j \right)^2,$$

$$\text{i.e., } \frac{\mathrm{d}}{\mathrm{d}t} \boldsymbol{\beta}_t = -\nabla_\beta \mathcal{L}(\boldsymbol{\beta}_t),$$

(22)

with $\theta_{j,t} = \lambda_j^{\frac{1}{2}} \beta_{j,t}$ as the estimate of $\theta_j^*$.

The Gaussian sequence dynamics in this subsection are understood coordinatewise, or as limits of truncations to finitely many coordinates. A finite truncation becomes essential in Section 4.3 when we introduce a learned orthogonal transformation of the noisy coordinates.

We hypothesize a strong equivalence between kernel gradient flow (20), which estimates $f^*$, and the Gaussian sequence gradient flow (22), which estimates $\{\theta_j^*\}_{j=1}^\infty$. We provide supporting evidence from the training dynamics and convergence rates in Sections 4.2.1 and 4.2.2.

### 4.2.1 Similar training dynamics

Under gradient flow, kernel regression and the Gaussian sequence model have similar training dynamics.

For kernel gradient flow, differentiating (20) gives

$$\frac{\mathrm{d}}{\mathrm{d}t} \boldsymbol{\beta}_t = -\nabla_\beta \mathcal{L}(\boldsymbol{\beta}_t) = -\frac{1}{n} \boldsymbol{\Lambda}^{\frac{1}{2}} \Psi(\boldsymbol{X}) \Psi(\boldsymbol{X})^\top \boldsymbol{\Lambda}^{\frac{1}{2}} \boldsymbol{\beta}_t + \frac{1}{n} \boldsymbol{\Lambda}^{\frac{1}{2}} \Psi(\boldsymbol{X}) \boldsymbol{Y}.$$

For large sample sizes, we use the rough approximations

- $\frac{1}{n} \Psi(\boldsymbol{X}) \Psi(\boldsymbol{X})^\top \approx \mathbf{I}$;

- $\left[ \frac{1}{n} \Psi(\boldsymbol{X}) \boldsymbol{Y} \right]_j \approx \lambda_j^{\frac{1}{2}} \beta_j^* + \tilde{\epsilon}_j, \quad \tilde{\epsilon}_j \overset{\text{i.i.d.}}{\sim} \mathcal{N}(0, n^{-1}).$

Accordingly, the gradient flow of $\boldsymbol{\beta}_t$ can be approximated by

$$\frac{\mathrm{d}}{\mathrm{d}t} \beta_{j,t} = -\lambda_j \beta_{j,t} + \lambda_j^{\frac{1}{2}} \left( \lambda_j^{\frac{1}{2}} \beta_j^* + \tilde{\epsilon}_j \right), \ \forall j \in \mathbb{N}^+.$$

(23)

In the Gaussian sequence model, (22) gives, for every $j = 1, 2, \ldots,$

$$\frac{\mathrm{d}}{\mathrm{d}t} \beta_{j,t} = -\lambda_j \beta_{j,t} + \lambda_j^{\frac{1}{2}} \left( \lambda_j^{\frac{1}{2}} \beta_j^* + \xi_j \right).$$

(24)

Because (21) assumes $\xi_j \overset{\text{i.i.d.}}{\sim} \mathcal{N}(0, n^{-1})$, (23) and (24) have the same form. Thus, the two gradient flows (20) and (22) are approximately the same.

### 4.2.2 Similar convergence rates

Section 2.3.2 gives convergence rates for kernel gradient flow under the eigenvalue decay rate and source condition assumptions. Under corresponding assumptions in the Gaussian sequence model, the gradient flow in (24), or equivalently (22), has the same generalization error convergence rates.

With zero initialization $\beta_{j,0} = 0$ for $j = 1, 2, \ldots$, solving (24) yields

$$\beta_{j,t} = \lambda_j^{-\frac{1}{2}}(1 - e^{-\lambda_j t})z_j.$$

The associated estimator is $\theta_{j,t} := \lambda_j^{\frac{1}{2}}\beta_{j,t} = (1 - e^{-\lambda_j t})z_j$. Writing $\boldsymbol{\theta}_t$ and $\boldsymbol{\theta}^*$ for the vectors of $\theta_{j,t}$ and $\theta_j^*$, respectively, its mean squared error under (21) is

$$\mathbb{E}\left\|\boldsymbol{\theta}_t - \boldsymbol{\theta}^*\right\|_2^2 = \sum_{j=1}^{\infty}(e^{-\lambda_j t}\theta_j^*)^2 + \frac{1}{n}\sum_{j=1}^{\infty}(1 - e^{-\lambda_j t})^2.$$

We impose the analogous eigenvalue decay rate (EDR), $\lambda_j \asymp j^{-\beta}$, and source condition, $\sum_{j=1}^{\infty}\lambda_j^{-s}(\theta_j^*)^2 < \infty$ and $\sum_{j=1}^{\infty}\lambda_j^{-r}(\theta_j^*)^2 = \infty$ for every $r > s$. Then, when $t = n^\alpha$ with $\alpha > 0$, the calculation gives (where $\beta$ denotes the EDR)

$$\mathbb{E}\left\|\boldsymbol{\theta}_t - \boldsymbol{\theta}^*\right\|_2^2 = \begin{cases} \Theta\left(n^{-s\alpha} + n^{-(1-\alpha/\beta)}\right), & \text{if } \alpha < \beta, \\ \Omega\left(1\right), & \text{if } \alpha \geq \beta. \end{cases}$$

These convergence rates coincide with the learning curve results for kernel gradient flow in (10).

## 4.3 Over-parameterized Gaussian sequence model

### 4.3.1 Applying the previous equivalence to different features/kernels

Section 4.2 discusses the hypothesized equivalence between kernel regression and the Gaussian sequence model for a fixed kernel. We now extend this equivalence to a family of kernels with different features. To make the feature change explicit, we retain $N$ coordinates, with eigenfunctions $\Psi(\boldsymbol{x}) = (\psi_1(\boldsymbol{x}), \ldots, \psi_N(\boldsymbol{x}))^\top$ and eigenvalue matrix $\boldsymbol{\Lambda} = \text{Diag}(\lambda_1, \ldots, \lambda_N)$.

Consider a restricted family of kernels obtained by rotating the feature coordinates and rescaling the eigenvalues relative to $\boldsymbol{\Lambda}$. For $\boldsymbol{A} \in O(N)$ and $\boldsymbol{D} = \text{Diag}(d_1, \ldots, d_N)$ with $d_j > 0$, define

$$\tilde{\Psi}(\boldsymbol{x}) = \boldsymbol{A}^\top\Psi(\boldsymbol{x}), \qquad \tilde{\boldsymbol{\Lambda}} = \boldsymbol{D}\boldsymbol{\Lambda}, \qquad \tilde{\Phi}(\boldsymbol{x}) = (\boldsymbol{D}\boldsymbol{\Lambda})^{\frac{1}{2}}\boldsymbol{A}^\top\Psi(\boldsymbol{x}).$$

The corresponding kernel is

$$\tilde{k}(\boldsymbol{x}, \boldsymbol{x}') = \Psi(\boldsymbol{x})^\top\boldsymbol{A}\boldsymbol{D}\boldsymbol{\Lambda}\boldsymbol{A}^\top\Psi(\boldsymbol{x}').$$

The truncated signal has coefficient vector $\tilde{\boldsymbol{\theta}}^* = \boldsymbol{A}^\top\boldsymbol{\theta}^*$ in the new orthonormal basis. For positive retained eigenvalues, the coefficient vector $\tilde{\boldsymbol{\beta}}^*$ in the new feature representation is determined by $\tilde{\boldsymbol{\theta}}^* = (\boldsymbol{D}\boldsymbol{\Lambda})^{1/2}\tilde{\boldsymbol{\beta}}^*$.

Kernel gradient flow for this new feature minimizes

$$\mathcal{L}(\tilde{\boldsymbol{\beta}}) = \frac{1}{2n}\left\|\boldsymbol{Y} - \tilde{\Psi}(\boldsymbol{X})^\top(\boldsymbol{D}\boldsymbol{\Lambda})^{\frac{1}{2}}\tilde{\boldsymbol{\beta}}\right\|_2^2. \tag{25}$$

Because $\tilde{\Psi}(\boldsymbol{X}) = \boldsymbol{A}^\top\Psi(\boldsymbol{X})$, the Gram matrix approximation in Section 4.2.1 is invariant under this rotation. Under the same finite-dimensional approximation, the flow in (25) has the following Gaussian sequence representation.

Let $\boldsymbol{Z} = \boldsymbol{\theta}^* + \boldsymbol{\xi}$, where $\boldsymbol{\theta}^* \in \mathbb{R}^N$ and $\boldsymbol{\xi} \sim \mathcal{N}(0, n^{-1}\mathbf{I}_N)$. Let $\tilde{\boldsymbol{\xi}} = \boldsymbol{A}^\top\boldsymbol{\xi}$ and $\tilde{\boldsymbol{Z}} = \tilde{\boldsymbol{\theta}}^* + \tilde{\boldsymbol{\xi}}$. Then $\tilde{\boldsymbol{\xi}} \sim \mathcal{N}(0, n^{-1}\mathbf{I}_N)$ and

$$\tilde{\boldsymbol{Z}} = \boldsymbol{A}^\top\boldsymbol{Z}. \tag{26}$$

Thus, the coordinate relation in the Gaussian sequence proxy is exact, whereas the connection from kernel gradient flow to this proxy still uses the approximation from Section 4.2.1. The Gaussian sequence loss for the new feature is

$$\tilde{\mathcal{L}}(\tilde{\boldsymbol{\beta}}) = \frac{1}{2} \left\| \tilde{\boldsymbol{Z}} - (\boldsymbol{D\Lambda})^{\frac{1}{2}} \tilde{\boldsymbol{\beta}} \right\|_2^2 \tag{27}$$

$$= \frac{1}{2} \left\| \boldsymbol{Z} - \boldsymbol{A}(\boldsymbol{D\Lambda})^{\frac{1}{2}} \tilde{\boldsymbol{\beta}} \right\|_2^2. \tag{28}$$

Within this finite-dimensional family, replacing the fixed kernel or feature in kernel gradient flow corresponds to the matrix $\boldsymbol{A}(\boldsymbol{D\Lambda})^{1/2}$ in the Gaussian sequence model.

### 4.3.2 Over-parameterized Gaussian sequence model

Building on this correspondence, we propose the following *over-parameterized Gaussian sequence model* as a prototype for the adaptive feature model in Definition 4.1.1.

**Definition 4.3.1** (Over-parameterized Gaussian sequence model). *Let $\boldsymbol{Z} = \boldsymbol{\theta}^* + \boldsymbol{\xi}$, where $\boldsymbol{\theta}^* \in \mathbb{R}^N$ and $\boldsymbol{\xi} \sim \mathcal{N}(0, n^{-1}\mathbf{I}_N)$. For $\boldsymbol{A}_t \in O(N)$, $\boldsymbol{s}_t \in \mathbb{R}^N$, and $\boldsymbol{\alpha}_t \in \mathbb{R}^N$, let*

$$\boldsymbol{D}_t = \mathrm{Diag}\left(e^{s_{1,t}}, \ldots, e^{s_{N,t}}\right)$$

*and define*

$$\hat{\boldsymbol{\theta}}_t = \boldsymbol{A}_t (\boldsymbol{D}_t \boldsymbol{\Lambda})^{\frac{1}{2}} \boldsymbol{\alpha}_t.$$

*Thus, $\boldsymbol{D}_t$ contains relative eigenvalue multipliers, while $(\boldsymbol{D}_t \boldsymbol{\Lambda})^{1/2}$ contains the associated feature scales. The parameters evolve to minimize*

$$\mathcal{L}(\boldsymbol{A}, \boldsymbol{s}, \boldsymbol{\alpha}) = \frac{1}{2} \left\| \boldsymbol{Z} - \boldsymbol{A}(\boldsymbol{D\Lambda})^{\frac{1}{2}} \boldsymbol{\alpha} \right\|_2^2,$$
$$\frac{\mathrm{d}}{\mathrm{d}t} \boldsymbol{A}_t = -\mathrm{grad}_{O(N)} \mathcal{L}, \qquad \frac{\mathrm{d}}{\mathrm{d}t} \boldsymbol{s}_t = -\nabla_{\boldsymbol{s}} \mathcal{L}, \qquad \frac{\mathrm{d}}{\mathrm{d}t} \boldsymbol{\alpha}_t = -\nabla_{\boldsymbol{\alpha}} \mathcal{L}, \tag{29}$$

*where $\mathrm{grad}_{O(N)}$ is the Riemannian gradient on the orthogonal group under the Frobenius metric. In this loss, $\boldsymbol{D} = \mathrm{Diag}(e^{s_1}, \ldots, e^{s_N})$. We initialize $\boldsymbol{A}_0 = \mathbf{I}_N$, $\boldsymbol{s}_0 = \mathbf{0}$, and $\boldsymbol{\alpha}_0 = \mathbf{0}$, so that $\boldsymbol{D}_0 = \mathbf{I}_N$ and the initial feature scale is $\boldsymbol{\Lambda}^{1/2}$.*

The model is a prototype for the adaptive feature model: $\boldsymbol{A}_t$ rotates feature coordinates and $\boldsymbol{D}_t$ changes their eigenvalue scales during training. The contribution here is the prototype model itself, rather than a new training mechanism for neural networks or a convergence or generalization theorem for the prototype. If $\boldsymbol{A}_t \equiv \boldsymbol{A}$ and $\boldsymbol{D}_t \equiv \boldsymbol{D}$, the loss reduces to (28) after setting $\boldsymbol{\alpha}_t = \tilde{\boldsymbol{\beta}}$. The fixed feature baseline is recovered by holding $\boldsymbol{A}_t \equiv \mathbf{I}_N$ and $\boldsymbol{s}_t \equiv \mathbf{0}$.

Relative to the vanilla gradient flow in (22), over-parameterization refers to the extra learnable parameters $\boldsymbol{A}_t$ and $\boldsymbol{s}_t$ in (29). These parameters provide a mechanism for adapting to latent structure in $\boldsymbol{\theta}^*$. They may improve generalization in suitable structures, but this is not a general guarantee and requires separate theoretical analysis.

We next describe the roles of $\boldsymbol{A}_t$ and $\boldsymbol{D}_t$. Let $\boldsymbol{u}_{j,t}$ be the $j$-th column of $\boldsymbol{A}_t$. Along an informative trajectory, the projections $\boldsymbol{u}_{j,t}^\top \boldsymbol{\theta}^*$ align with directions having larger effective eigenvalues $d_{j,t}\lambda_j$, while directions with negligible projections have smaller effective eigenvalues. Thus, $\boldsymbol{A}_t$ and $\boldsymbol{D}_t$ adjust eigenvectors and eigenvalues, respectively.

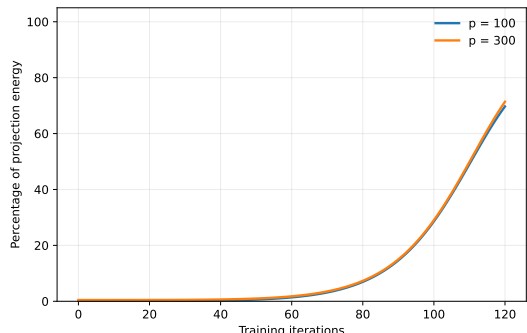

Figure 6: $x$-axis: number of training iterations; $y$-axis: the percentage of the first $p$ projections: $\sum_{j=1}^{p} f_{j,t}^2 / \sum_{j=1}^{N} f_{j,t}^2$, $p = 100, 300, N = 500$. The projection energy of $f_{j,t}$ increasingly concentrates on directions with the largest effective eigenvalues during training.

### 4.3.3 Simulation

We simulate the over-parameterized Gaussian sequence model in Definition 4.3.1. Set $\boldsymbol{s}_0 = \boldsymbol{0}$ and $\boldsymbol{A}_0 = \mathbf{I}_N$, so that $\boldsymbol{D}_0 = \mathbf{I}_N$. To approximate (29), we use Riemannian gradient descent with a polar retraction after each update of $\boldsymbol{A}_t$. The vectors $\boldsymbol{s}_t$ and $\boldsymbol{\alpha}_t$ are updated by ordinary gradient descent. We use $N = 500$, $n = 4000$, and a learning rate of 0.5. The signal and eigenvalues are $\theta_j^* = 1/(N - j + 2)$ and $\lambda_j = 1/(j+5)^2$ for $j = 1, \ldots, N$. Thus, $\{\lambda_j\}_{j=1}^N$ decreases while $\{\theta_j^*\}_{j=1}^N$ increases.

For each $j = 1, 2, \ldots, N$, let $j'$ index the $j$-th largest effective eigenvalue $d_{j',t} \lambda_{j'}$, and let $\boldsymbol{u}_{j',t}$ be the corresponding column of $\boldsymbol{A}_t$. Let $f_{j,t} := \boldsymbol{u}_{j',t}^\top \boldsymbol{\theta}^*$. Then $f_{j,t}$ is the projection onto the eigenvector corresponding to the $j$-th largest effective eigenvalue.

At initialization, $\boldsymbol{A}_0 = \mathbf{I}_N$, $\boldsymbol{D}_0 = \mathbf{I}_N$, and $f_{j,0} = \theta_j^*$, so the data are misaligned with the feature: the coefficients $\theta_j^*$ are smaller in directions with larger $\lambda_j$. The quantities $f_{j,t}$ are oracle diagnostics in this simulation because they use the known vector $\boldsymbol{\theta}^*$. Figure 6 shows that the percentage of the first $p$ projections, $\sum_{j=1}^{p} f_{j,t}^2 / \sum_{j=1}^{N} f_{j,t}^2$ for $p = 100$ and 300, increases with the number of training iterations.

## 4.4 Over-parameterized models in related literature

Li and Lin (2024) studied a data-generating model similar to that in Definition 4.3.1:

$$z_j = \theta_j^* + \xi_j, \quad j = 1, 2, \ldots,$$

where the noise variables $\{\xi_j\}_{j=1}^\infty$ are $\epsilon^2$-sub-Gaussian. They estimated $\theta_j^*$ by $\theta_{j,t} := a_{j,t} \beta_{j,t}$ and applied gradient flow to $\boldsymbol{a}_t = (a_{1,t}, a_{2,t}, \ldots)^\top$ and $\boldsymbol{\beta}_t = (\beta_{1,t}, \beta_{2,t}, \ldots)^\top$ to minimize

$$\mathcal{L}(\boldsymbol{a}, \boldsymbol{\beta}) = \frac{1}{2} \sum_{j=1}^\infty (z_j - a_j \beta_j)^2. \tag{30}$$

They initialized $a_{j,0} = \lambda_j^{\frac{1}{2}}$ for a sequence $\lambda_j \asymp j^{-\beta}$ with $\beta > 1$ and set $\beta_{j,0} = 0$ for $j = 1, 2, \ldots$. Let $\boldsymbol{\theta}_t$ and $\boldsymbol{\theta}^*$ denote the vectors of $\theta_{j,t}$ and $\theta_j^*$, respectively.

They also compared the generalization error of the gradient flow (30) with that of the vanilla gradient flow (22), in which $\boldsymbol{a}_t$ remains fixed at $a_{j,t} \equiv \lambda_j^{\frac{1}{2}}$ for $j = 1, 2, \ldots$. Corollary 3.3 of Li and

Lin (2024) provides a concrete example where the gradient flow (30) has a faster generalization error convergence rate than the vanilla gradient flow (22).

For a finite truncation with $a_{j,t} \geq 0$ and $\lambda_j > 0$, setting $\boldsymbol{A}_t \equiv \mathbf{I}_N$ and choosing $d_{j,t} = a_{j,t}^2/\lambda_j$ gives $(\boldsymbol{D}_t \boldsymbol{\Lambda})^{1/2} = \mathrm{Diag}\{a_{1,t}, \ldots, a_{N,t}\}$. Thus, (30) is the counterpart of Definition 4.3.1 with only diagonal rescaling, whereas our prototype also permits rotations of the feature coordinates. The rotational extension may improve generalization in suitable structures, but a general comparison remains an open theoretical question.

A separate line of research studies the advantages of over-parameterization in high-dimensional linear regression with sparse signals (Li et al., 2021; Vaskevicius et al., 2019; Zhao et al., 2022). For example, Zhao et al. (2022) applied gradient descent to $\boldsymbol{g}_t, \boldsymbol{l}_t \in \mathbb{R}^d$ to minimize the square loss

$$\mathcal{L}(\boldsymbol{g}, \boldsymbol{l}) = \frac{1}{2n}\|\boldsymbol{Y} - \boldsymbol{X}^\top(\boldsymbol{g} \circ \boldsymbol{l})\|_2^2,$$

where "$\circ$" denotes the Hadamard product. They showed that the over-parameterized version of gradient descent could lead to a near-minimax optimal rate and a faster dimension-free convergence rate in the strong signal case.

# 5    Discussion

Theoretically analyzing the feature learning characteristics of neural network models is a challenging task. The primary difficulty lies in the complex training dynamics of these models. Although existing results have made valuable progress (Section 3.3), fully understanding feature learning and its benefits for generalization ability still requires considerable effort. This paper introduces the over-parameterized Gaussian sequence model (Definition 4.3.1) as a prototype for the adaptive feature model. This prototype borrows the concept of a good feature from fixed kernel regression theory and retains the rotation and rescaling mechanisms of adaptive feature learners. Relative to Lecué et al. (2025) and Gavrilopoulos et al. (2024), the contribution is this prototype model rather than a new general definition of feature learning, a training mechanism for neural networks, or a mathematical guarantee.

We expect that the over-parameterized Gaussian sequence model can offer insights into how feature learning occurs and may improve generalization in suitable structures. As a prototype, while it does not by itself transfer results to real neural networks, we expect to establish such a connection in the future.   To this end, we need to investigate (but not limited to) the following questions in the future:

- Theoretical understanding of the over-parameterized Gaussian sequence model in Definition 4.3.1. Specifically, the first step is to characterize the alignment between $\boldsymbol{A}_t, \boldsymbol{D}_t$ and the true parameters as the gradient flow progresses. The next step is to analyze whether the alignment improves generalization ability relative to vanilla gradient flow or models without over-parameterization. We can begin with a simple family of feature maps and gradually increase their complexity. For example, Li and Lin (2024) considered the simplest case where the eigenvalues are adjustable. In addition, one can consider the case where $\boldsymbol{A}_t$ varies among a family of orthogonal matrices, making the eigenvectors adjustable.

- Rigorous equivalence between the (adaptive feature) kernel regression model and the (over-parameterized) Gaussian sequence model, especially in the high dimensions. We have shown some evidence of the equivalence in Section 4.2.2, but the rigorous proof and the high-dimensional case remain open problems. Establishing this equivalence is required before conclusions from the prototype can be extended to the adaptive feature model or real neural networks.

- The specific architecture of neural networks (e.g., the number of layers, fully connected or convolutional neural networks, etc.) will inevitably affect the manner and efficiency of feature learning. Characterizing the differences between various neural network architectures using the adaptive feature model will be a more advanced problem.

## Experiments details

We provide some details of the neural networks experiments in Section 4.1.

- The fully connected neural network (for MNIST). Input dimension = 784; first hidden layer dimension = 500; second hidden layer dimension = 500; readout layer dimension = 1.

- The convolutional neural network (for CIFAR-10). Input shape = $3 * 32 * 32$; first convolutional layer: 64 channels, $5 * 5$ kernel, dropout = 0.5, max pooling; second convolutional layer: 256 channels, $5 * 5$ kernel, dropout = 0.5, average pooling; third convolutional layer: 32 channels, $5 * 5$ kernel, dropout = 0.5, no pooling; a fully connected layer: dimension=500; readout layer: dimension = 1.

We use ReLU activation, MSE loss, batch size of 32 and a learning rate of 0.001. We train neural networks in all 60000 samples of MNIST or CIFAR-10 and use 10000 of these samples to calculate $f_j := \frac{1}{\sqrt{n}} \hat{\boldsymbol{v}}_j^\top \boldsymbol{Y}$ (i.e., $n = 10000$) in Section 4.1.

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
