# OpenReview forum: "Towards a Statistical Understanding of Neural Networks: Beyond the Neural Tangent Kernel Theories"
_SLADS/Section_C — Under review for SLADS_Section_C_

### Review · Reviewer_hGCj · 2026-07-03

**Summary Of Contributions:**

The paper aims to develop a statistical perspective on neural networks beyond the classical NTK framework, with a particular emphasis on generalization and feature learning. In my understanding, the main contributions of the paper can be summarized as follows.

First, the paper revisits the connection between kernel regression and the Gaussian sequence model from the perspective of gradient flow. In particular, it argues that, under gradient flow, the kernel regression model and the Gaussian sequence model have analogous training dynamics and share the same convergence rates for the generalization error. This viewpoint is used to motivate the Gaussian sequence model as a simplified statistical proxy for studying the behavior of kernel-based learning methods, and serves as a starting point for moving beyond fixed-kernel theories.

Second, building on this connection, the paper introduces the concept of an over-parameterized Gaussian sequence model. The motivation is to go beyond the standard Gaussian sequence model, which mainly corresponds to a fixed-feature setting, and to incorporate a form of adaptive feature learning into the model. To this end, the paper augments the classical sequence model with additional trainable quantities, such as A_t and D_t, with the goal of mimicking how neural networks adapt their representations during training. In this sense, the over-parameterized Gaussian sequence model is proposed as a prototype statistical model for studying alignment, feature learning, and their possible implications for neural network generalization.

**Audience:**

Yes

**Broader Impact Concerns:**

This paper is theoretical in nature and does not raise direct ethical or societal concerns. Any broader impact is indirect and typical of foundational machine learning research.

**Claims And Evidence:**

Yes

**Requested Changes:**

The required changes are listed below.

1.	The paper should provide a clearer explanation of how NTK theory establishes the connection between neural networks and kernel gradient flow, including
(1)	what the neural tangent kernel is and how it is defined;
(2)	how the NTK arises from the linearization of neural networks around initialization;
(3)	why, in the infinite-width or sufficiently wide regime, the network training dynamics can be approximated by a kernel method;

2.	The paper should more clearly clarify the relationship between kernel regression including kernel ridge regression and kernel interpolation, and neural networks under the NTK framework. In particular, the paper should more clearly explain in what sense does NTK reduce neural network training to a kernel regression problem, and whether the neural network has the same generalization behavior as kernel regression when analyzed in the NTK regime.

3.	The paper should discuss more how its review differs from or improves upon existing papers that review or analyze NTK, for example,
[1] Seleznova, M., & Kutyniok, G. (2022, April). Analyzing finite neural networks: Can we trust neural tangent kernel theory?. In Mathematical and Scientific Machine Learning (pp. 868-895). PMLR.
[2] Golikov, E., Pokonechnyy, E., & Korviakov, V. (2022). Neural tangent kernel: A survey. arXiv preprint arXiv:2208.13614.
[3] Jiang, K., Cohen, J., & Li, Y. (2026). Understanding the evolution of the neural tangent kernel at the edge of stability. Advances in Neural Information Processing Systems, 38, 31996-32036.
In particular, paper [3] also discusses alignment, feature learning, and generalization, which appear closely related to the themes of the current paper. Therefore, the paper should explain more clearly what the main difference is between the current paper and paper [3], and what exactly is unique of the present paper.

5.	For the proposed over-parameterized Gaussian sequence model, the paper should develop a more rigorous analysis, including
(1)	the convergence behavior of \theta^* in the over-parameterized Gaussian sequence model, analogous to the analysis available for the classical Gaussian sequence model;
(2) how the convergence behavior or estimation error in this model can shed light on the generalization error of neural networks.


Below, I list additional minor issues that would further improve the clarity and presentation of the paper.
1. On page 1, \mu should refer to the joint distribution on \mathcal{X}, since \mathcal{X} \subset \mathbb{R}^d. A similar issue also appears in the statement on page 8: “Assume that \mathcal{X} \subset \mathbb{R}^d is a bounded domain with a smooth boundary and the marginal distribution \mu on \mathcal{X} has Lebesgue density 0<c\le p(x)\le C for two constants c and C.”

2. In Eq. (1) on page 1, is the noise term \epsilon assumed to be independent of \mathbf{x}?

3. Some section and subsection titles are not sufficiently formal or clear. For example, the use of “&” in Section 2 is informal, and the phrase “kernel/feature” in Section 3 is ambiguous. In addition, Subsections 3.2.1 and 3.2.2 should also be given titles.

4. Some notation and formulas are not clearly explained. For example:
   What is the precise meaning of (a_j)_{j=1}^{\infty} \in \ell^2 on page 4?
   What does the arrow mean in [\mathcal{H}]^{s_2} \rightarrow [\mathcal{H}]^{s_1} \rightarrow [\mathcal{H}]^{0} on page 6?

5. The two subfigures in Figure 1 should be labeled explicitly as (a) and (b).

6. On page 13, in the formula k(\bm x, \bm x^{\prime})=\Psi(\bm x)^{\top}\bm{\Lambda}\Psi(\bm x), the second \Psi(\bm x) should presumably be \Psi(\bm x^{\prime}).

7. The sentence “the projections \langle f^*, \psi_j\rangle_{L^2} concentrate more on the eigenspaces corresponding to larger eigenvalues (\lambda_j)” is not very clearly written and should be rephrased for clarity.

8. On page 22, it is unclear how the formula \tilde{\bm Z}=\bm A^{\top}\bm Z is obtained. More explanation or derivation would be helpful.

9. In Eq. (27) on page 23, it seems that \bm D^{1/2} should be used instead.

10. The value of D_0 given in the second line of Subsection 4.3.3 is inconsistent with the value stated in the ninth line on page 24.

11. The motivation for introducing A_t and D_t into the Gaussian sequence model is not sufficiently explained. It would be helpful to clarify what advantages they bring compared with the standard Gaussian sequence model.

**Strengths And Weaknesses:**

Strength:
1. The paper provides a broad and detailed review of the generalization behavior of kernel regression, covering both the fixed-dimensional and high-dimensional settings. In particular, the discussion of convergence rates of kernel ridge regression, kernel interpolation, and related statistical results is extensive and, in many places, clearly presented.
2. The paper gives a useful discussion of why the classical NTK framework is limited in its ability to explain feature learning in neural networks. The motivation for moving beyond fixed-kernel theories is well articulated, and the paper identifies feature adaptation and alignment as important phenomena that are not adequately captured by standard NTK analyses.
3. The introduction of the adaptive feature viewpoint and the over-parameterized Gaussian sequence model reflects an interesting attempt to develop a more flexible statistical framework for understanding neural networks beyond NTK-type kernel theories.

Weaknesses:
1. Although the paper discusses the generalization behavior of kernel regression in considerable detail, I found that the main focus of the review part is often on kernel regression itself rather than on neural networks. In particular, the paper does not sufficiently connect the kernel-regression results back to the original neural network problem. Since the stated goal is to improve the statistical understanding of neural networks, the paper should more clearly explain what the reviewed kernel-regression results imply for neural network training and generalization under the NTK framework, and what conclusions should ultimately be drawn about neural networks rather than kernel methods alone.
2. The current analysis of the over-parameterized Gaussian sequence model is not sufficiently developed. At present, the paper mainly introduces the model and provides heuristic motivation, but it does not establish substantive theoretical results on its optimization dynamics, convergence behavior, or statistical performance. As a result, it is difficult to assess the extent to which this model truly advances the theoretical understanding of neural-network feature learning. The implications of the proposed Gaussian sequence model for neural network theory are not yet demonstrated.

---

> ### Author Response · Authors · 2026-07-15
> **Response to Reviewer hGCj**
>
> ## Overall response
>
> We thank the reviewer for identifying the need to connect the kernel regression review more explicitly to neural networks and to state the scope of the over-parameterized Gaussian sequence model more carefully.  We have added a bridge from linearization to NTK and KGF, clarified the relation among KGF, kernel interpolation, and KRR, and expanded the introduction's positioning against the requested survey and alignment literature.  We have also corrected the rotation, scale, and initialization details of the Gaussian sequence model and now present it explicitly as a prototype for the adaptive feature model.
>
> ## Point-by-point responses to the requested changes
>
> ### 1–2. The NTK reduction and connection back to neural networks
>
> We define the NTK as the inner product kernel of tangent features from the first order linearization at initialization and state the sufficiently wide approximation to kernel gradient flow under the cited result's initialization, width, and continuous time assumptions.  Under that result, both the network predictor and its risk are approximated by the corresponding KGF predictor and risk, so the KGF generalization conclusions apply only in the stated regime.  We further distinguish finite time KGF from kernel interpolation, which arises as $t\to\infty$ under strict positive definiteness of the empirical kernel matrix and agrees with the $\lambda\to0$ limit of KRR; the relation between KGF and KRR is stated at the rate level.  See Sections 2.1 and 2.2.
>
> ### 3. Distinct contribution
>
> We added a contribution and related work discussion in the introduction.  It distinguishes the survey's organization around the transition from fixed features to learned representations from the NTK accounts by Seleznova and Kutyniok and by Golikov, Pokonechnyy, and Korviakov, and positions Jiang, Cohen, and Li as a study of NTK eigenvector alignment at the edge of stability for a specified training algorithm.  See the Introduction (pp. 2 to 3).
>
> ### 4. Over-parameterized Gaussian sequence model
>
> We agree that convergence and the connection to neural network generalization are important open questions.  In this revision, we make the rotation and scale dynamics explicit, correct the transformed coordinate representation, and present the model as a prototype for the adaptive feature model.  A full treatment would require a separate analysis of the coupled nonconvex dynamics and a rigorous link from the prototype to neural network generalization, which is beyond the scope of the present survey.  We have therefore narrowed the manuscript's claims and state this limitation explicitly: we do not claim a new convergence theorem or a transfer result to neural networks.  See Sections 4.2 and 4.3, Definition 4.3.1, and the Discussion.
>
> ## Point-by-point responses to the minor issues
>
> ### 1. Input distribution
>
> We define $\mu$ as the input distribution on $\mathcal X$ in the model setup and use the same terminology in the Sobolev example.  See the Introduction and Section 2.3.
>
> ### 2. Noise independence
>
> We now state that the noise in Equation (1) is independent of the input.  See the Introduction.
>
> ### 3. Headings and subsection titles
>
> We replaced the ambiguous headings and added descriptive titles to Sections 3.2.1 and 3.2.2.  See Sections 2 and 3.
>
> ### 4. Sequence and embedding notation
>
> We define $\ell^2$ as the space of square summable sequences and explain that $\hookrightarrow$ denotes a continuous embedding.  See Section 2.2.
>
> ### 5. Figure 1 panels
>
> We added the panel labels (a) and (b) to Figure 1.
>
> ### 6. Kernel expressions
>
> We corrected the second feature map argument in the Mercer representation and the empirical NTK matrix entry.  See Sections 3.1 and 3.2.
>
> ### 7. Alignment wording
>
> We rewrote the alignment definition in terms of squared projection energy in eigenspaces associated with larger eigenvalues.  See Section 3.2.
>
> ### 8. Rotated observation
>
> We define the rotated noise before deriving the transformed observation, so that $\widetilde{\mathbf{Z}}=\mathbf{A}^{\top}\mathbf{Z}$ follows directly from orthogonal invariance.  See Section 4.3.1.
>
> ### 9. Feature scale
>
> We now write the transformed feature as $(\mathbf{D}\boldsymbol{\Lambda})^{1/2}\mathbf{A}^{\top}\Psi(\mathbf{x})$ and use the same square root scale in the Gaussian sequence loss and model definition.  See Section 4.3.1 and Definition 4.3.1.

---

> > ### Author Response · Authors · 2026-07-15
> >
> > ### 10. Initialization of $\mathbf{D}$
> >
> > We initialize $\mathbf{A}_0=\mathbf{I}_N$, $\boldsymbol{s}_0=\boldsymbol{0}$, and $\boldsymbol{\alpha}_0=\boldsymbol{0}$, which gives $\mathbf{D}_0=\mathbf{I}_N$ and initial feature scale $\boldsymbol{\Lambda}^{1/2}$.  The simulation uses the same initialization.  See Definition 4.3.1 and Section 4.3.3.
> >
> > ### 11. Motivation for $\mathbf{A}_t$ and $\mathbf{D}_t$
> >
> > We now explain that $\mathbf{A}_t$ rotates the feature coordinates while $\mathbf{D}_t$ changes their relative eigenvalue scales.  Holding $\mathbf{A}_t=\mathbf{I}_N$ and $\mathbf{D}_t=\mathbf{I}_N$ recovers the fixed feature baseline, whereas the additional parameters provide a mechanism for adapting the representation to latent structure.  We state that this mechanism may improve generalization in suitable structures but is not a general guarantee.  See Definition 4.3.1 and Section 4.3.2.

---

### Review · Reviewer_3FsL · 2026-07-05

**Summary Of Contributions:**

This paper surveys several aspects of neural network theory, with a particular emphasis on the transition from kernel regression, lazy learning, and NTK-type analyses to feature learning. The submission discusses several representative theoretical frameworks and aims to organize recent developments around the role of learned features, adaptive representations, and kernelized descriptions of neural networks. The topic is important and timely. A survey clarifying the relation between kernel methods, neural networks, adaptive kernels, and feature learning would be valuable for the community. However, in its current form, the paper still requires substantial revision.

**Audience:**

Yes

**Broader Impact Concerns:**

I do not have specific broader impact concerns. The paper is a theoretical survey, and I do not see immediate ethical or societal risks requiring additional discussion beyond the standard broader impact statement.

**Claims And Evidence:**

Yes

**Requested Changes:**

I recommend major revision. The following changes are important for improving the paper.

1. The authors should systematically distinguish algorithms from estimators throughout the paper. This should be one of the organizing principles of the survey. The paper should clearly explain that studying the statistical properties of a theoretically defined estimator is different from proving that a concrete training algorithm converges to an estimator with those properties.

2. The discussion of algorithm-independent and algorithm-dependent theories should be substantially expanded. The authors should explain why algorithm-dependent theory is needed in neural network theory. For example, Schmidt-Hieber (2017) and Suzuki (2018) analyze estimators, while the convergence of practical training algorithms to such estimators is a separate and generally unresolved issue. The authors should also explain that different algorithms, and even different initializations of the same algorithm, may lead to estimators with substantially different statistical properties.

3. The authors should reformulate the discussion of the major challenge in studying neural networks in terms of the empirical risk landscape. For a neural network class $\mathcal F_{\mathrm{NN}}$, the central difficulty is the nonconvexity of

$$
f\in\mathcal F_{\mathrm{NN}}\mapsto P_N\ell_f .
$$

The paper should explain how this nonconvexity obstructs general guarantees that an actual training algorithm converges to a theoretically defined estimator.

4. The discussion of NTK should be corrected and made more precise. NTK/lazy training should be explained in terms of the relation between width, scaling, and the step size of the training algorithm. The paper should cite and discuss Chizat, Oyallon, and Bach, “On lazy training in differentiable programming.” The authors should also clarify that infinite width alone does not imply lazy behavior. Mean-field limits provide infinite-width regimes with feature learning; see Yang and Hu, “Tensor Programs IV: Feature Learning in Infinite-Width Neural Networks.”

5. The authors should clarify the role of kernelization. In the NTK regime, one can view the neural network class as being kernelized into a function class $\mathcal F_{\mathrm{NTK}}$, for which the corresponding landscape

$$
f\in\mathcal F_{\mathrm{NTK}}\mapsto P_N\ell_f
$$

is convex. This convexity is what makes kernel gradient flow amenable to convergence analysis. The authors may also refer to Boyer, “Living la vida loca: learning in interpolation regimes,” for related perspectives.

6. The transition from kernel regression to feature learning should be made more convincing. The authors should explain in what statistical regimes kernel regression is suboptimal compared with algorithms that perform feature learning. They should also explain why this suboptimality occurs. This is necessary because, in many classical nonparametric regression problems, kernel regression or spectral methods already achieve minimax optimality.

7. The discussion of lazy learning versus feature learning should include the literature on the information exponent. In particular, the authors should discuss Abbe, Boix-Adsera, and Misiakiewicz, “SGD learning on neural networks: leap complexity and saddle-to-saddle dynamics,” and Ben Arous, Gheissari, and Jagannath, “High-dimensional limit theorems for SGD: Effective dynamics and critical scaling.” This would make the discussion in Section 3.2, especially the claim that “the alignment between the true function and the chosen feature matters,” more convincing.

8. The authors should expand the discussion of alignment between the target function and the chosen feature representation. In particular, they should compare their discussion with Lecué, Li, and Shang, “Sharp convergence rates for Spectral methods via the feature space decomposition method.” That work studies sharp convergence rates for spectral methods through a feature-space decomposition, and it explicitly quantifies how the target function aligns with the relevant spectral directions. Since the present survey also emphasizes that the chosen feature representation matters, a comparison with this feature-space decomposition viewpoint would help clarify the precise relation between kernel/spectral methods and feature learning.

9. The authors should clarify whether the alignment phenomena discussed in Section 3.2 are meant to describe a limitation of fixed-feature methods, a mechanism through which adaptive-feature algorithms improve over fixed kernels, or both. This distinction is important because alignment can already play a decisive role in fixed-kernel spectral methods, while feature learning concerns the algorithmic production of a more favorable feature representation.

10. The authors should add subtitles for Sections 3.2.1 and 3.2.2. In the current version, these parts look like possible compilation errors, which makes the structure of the section unclear.

11. Figure 4 should be revised or explained in detail. The authors should clarify why the displayed inclusions hold, especially why adaptive features contain neural networks. If strict inclusions are intended, the authors should justify the strictness of each inclusion.

12. The adaptive-kernel discussion should cite and compare with the two recursive kernel machine papers by Radhakrishnan, Beaglehole, Pandit, and Belkin. The relation between the adaptive-kernel viewpoint in the present survey and recursive kernel machines should be explicitly discussed.

Overall, the paper addresses an important topic, but as a survey it needs substantial revision in literature coverage, conceptual clarity, and justification of several key claims.

**Strengths And Weaknesses:**

Strengths:

1. The paper addresses an important and timely topic. Understanding the relation between kernel regression, NTK/lazy training, adaptive kernels, and feature learning is central to the current theory of neural networks.

2. The paper attempts to connect several different viewpoints, including kernel methods, learned representations, and adaptive feature maps. This could be useful for readers who want a broad overview of this area.

3. The discussion around lazy learning versus feature learning has the potential to become a valuable part of the survey, provided that the relevant literature and conceptual distinctions are developed more carefully.

Weaknesses:

1. The literature coverage is incomplete for a survey. Several important lines of work related to feature learning, information exponent, adaptive kernels, algorithm-dependent statistical theory, and feature-space alignment are missing or insufficiently discussed.

2. The distinction between an algorithm and an estimator is not sufficiently emphasized. This distinction should be central to the survey. For example, some existing works study statistical properties of certain neural network estimators, while leaving open whether a practical training algorithm converges to those estimators. This distinction has major implications for neural network theory.

3. The discussion of algorithm-independent versus algorithm-dependent theory needs to be expanded. For example, Schmidt-Hieber (2017) and Suzuki (2018) study statistical properties of certain neural network estimators. However, these works do not by themselves provide guarantees that a concrete training algorithm converges to these estimators. Different training algorithms, and even different initializations of the same training algorithm, may lead to estimators with substantially different statistical properties. This makes algorithm-dependent statistical theory an essential issue in neural network theory.

4. The paper should give a more precise account of the main optimization challenge in neural networks. For a neural network class $\mathcal F_{\mathrm{NN}}$, a central difficulty is the highly nonconvex empirical risk landscape

$$
f\in\mathcal F_{\mathrm{NN}}\mapsto P_N\ell_f .
$$

This nonconvexity is precisely what makes it difficult to prove that a practical training algorithm converges to the theoretically defined estimator.

5. The discussion of NTK and lazy training is too imprecise. NTK, or more generally the lazy regime, should not simply be described as the statement that sufficiently wide neural networks have training dynamics well approximated by kernel gradient flow. The lazy regime depends on the relation between the step size of the training algorithm, the scaling of the neural network, and the width. The authors should discuss this point more carefully and may refer to Chizat, Oyallon, and Bach, “On lazy training in differentiable programming.” Mean-field neural networks also arise as infinite-width limits, but they can still have feature learning properties; see Yang and Hu, “Tensor Programs IV: Feature Learning in Infinite-Width Neural Networks.”

6. The connection from kernel regression to feature learning is not yet fully convincing. The authors should explain when kernel regression has worse statistical performance than algorithms with feature learning capability, and why this happens. In classical nonparametric regression problems, kernel regression or spectral methods can already attain minimax optimality, so feature learning cannot improve the statistical rate in those cases. The advantage of feature learning therefore needs to be tied to more precise structural properties of the problem.

7. The current discussion of polynomial barriers is useful but incomplete. The authors omit an important line of work centered around the information exponent, including Abbe, Boix-Adsera, and Misiakiewicz, “SGD learning on neural networks: leap complexity and saddle-to-saddle dynamics,” and Ben Arous, Gheissari, and Jagannath, “High-dimensional limit theorems for SGD: Effective dynamics and critical scaling.” These works are related to, but distinct from, the polynomial barrier perspective. The polynomial barrier still concerns alignment between the target function and feature directions, while the information-exponent viewpoint emphasizes the vanishing of the projection of the target function onto $P_{\leq \mathrm{IE}}$.

8. The paper repeatedly uses the idea that alignment between the target function and the chosen feature representation matters, but this point is not sufficiently connected to the existing literature. In particular, the authors should compare their discussion with Lecué, Li, and Shang, “Sharp convergence rates for Spectral methods via the feature space decomposition method.” That work studies sharp convergence rates for spectral methods by decomposing the feature space and quantifying how the target function aligns with the relevant spectral directions. Such a comparison would clarify what is new in the present survey’s viewpoint, and would also make the statement in Section 3.2 that “the alignment between the true function and the chosen feature matters” more convincing.

9. Figure 4 requires further explanation. The figure appears to suggest several strict inclusion relations. The authors should explain why these inclusions hold, especially why adaptive features contain neural networks. If strict inclusions are intended, the authors should explain why each inclusion is strict.

10. The adaptive-kernel viewpoint seems closely related to the recursive kernel machine papers by Radhakrishnan, Beaglehole, Pandit, and Belkin. These papers should be cited and compared in the adaptive-kernel discussion.

---

> ### Author Response · Authors · 2026-07-15
> **Response to Reviewer 3FsL**
>
> ## Overall response
>
> We thank the reviewer for the detailed suggestions on the scope and exposition of the survey.  In response, we have sharpened the distinction between guarantees for theoretically defined estimators and guarantees for specified training dynamics; clarified the NTK reduction and the regimes in which feature learning can persist; and expanded the discussion of fixed feature limitations, alignment, and adaptive kernels.  We have also revised the organization of Section 3 and replaced Figure 4 with a conceptual map that makes no inclusion or strictness claim.
>
> ## Point-by-point responses to the requested changes
>
> ### 1–2. Algorithm versus estimator
>
> We restructured the introduction around the distinction between generalization guarantees for theoretically defined estimators and guarantees for specified training dynamics.  The revised discussion cites the published version, Schmidt-Hieber (2020), of the 2017 preprint referenced by the reviewer, together with Suzuki (2018), as examples at the estimator level and carries the distinction into the kernel regression and feature learning reviews.  We also state that different training procedures and initializations can lead to different parameter trajectories and predictors.  See the Introduction (p. 2), Section 2.1, and Section 3.3.
>
> ### 3. Optimization landscape
>
> We now state the parameterized empirical risk problem and explain that its nonconvexity prevents a guarantee at the estimator level from establishing reachability by a practical training procedure.  See the Introduction (p. 2) and Section 2.1.
>
> ### 4. NTK and lazy training
>
> Section 3.3 now states that, even at infinite width, the parameterization, initialization scale, and learning rate scaling jointly determine whether training remains in a kernel regime or exhibits feature learning.  We cite Chizat, Oyallon, and Bach, and Yang and Hu, and note that mean field limits can exhibit increasing alignment during training.  See Sections 2.1 and 3.3.
>
> ### 5. Kernelization and its limit
>
> We added a local bridge from the first order linearization at initialization to the NTK and its kernel gradient flow approximation, and explain that the resulting kernelized training problem is convex.  We also distinguish finite time kernel gradient flow from kernel interpolation and state the relation between KGF and KRR at the level of regularization rates.  See Sections 2.1 and 2.2.
>
> ### 6. Fixed features and feature learning
>
> We now distinguish the matched classical regime, where an appropriately regularized fixed kernel method can attain the minimax rate, from structural representation mismatch.  The polynomial approximation barrier illustrates the latter situation; feature learning is described as potentially advantageous only when training changes the representation to improve alignment with the target.  See Section 3.1 (p. 14).
>
> ### 7–9. Alignment and information exponent
>
> We added a direct comparison paragraph in Section 3.2.  It separates the limitation of alignment in a representation fixed before training from the mechanism by which training can obtain a more favorable representation, and relates this distinction to the requested information exponent and feature space decomposition work.  See Section 3.2 (p. 15).
>
> ### 10–11. Structure and Figure 4
>
> We added descriptive titles to the two examples in Section 3.2.  We also replaced Figure 4 with a conceptual map: neural network training is abstracted as an adaptive feature model, the NTK approximation in the lazy regime gives a fixed feature approximation, and the over-parameterized Gaussian sequence model is a prototype for the adaptive feature model.  The revised figure makes no inclusion or strictness claim.  See Section 3.2 and Figure 4 in Section 4.1 (p. 19).
>
> ### 12. Adaptive kernels
>
> We added the two recursive feature machine references and explain that their construction updates the feature map from the data, in contrast to the fixed feature map in an NTK approximation.  See Section 3.3 (p. 16).

---

> > ### Comment · Reviewer_3FsL · 2026-07-20
> >
> > Thank you for the substantial revision. The discussion of the estimator–algorithm distinction, the NTK regime, and the limitations of fixed representations has been significantly improved. One important issue from my previous report, however, still seems insufficiently addressed.
> >
> > The revised manuscript now cites Lecué et al. (2025) in connection with the distinction between alignment for a fixed representation and feature learning. However, it does not yet explain how the conceptual framework proposed in the present manuscript differs from the formal treatment of alignment and feature learning developed in that work.
> >
> > Similarly, Gavrilopoulos et al. (2024) is still presented mainly as part of the general kernel-regression literature, although that work also studies feature learning generated by specified training dynamics through a data-dependent conjugate kernel, together with the resulting change in alignment and statistical performance. Consequently, the distinction emphasized in the response between guarantees for theoretically defined estimators and guarantees for specified training dynamics does not, by itself, separate the present manuscript from that work.
> >
> > I therefore ask the authors to provide a direct and substantive comparison between these works and the adaptive feature model and over-parameterized Gaussian sequence model proposed here. In particular, the manuscript should state precisely which component of the claimed “new paradigm” is new: the conceptual abstraction, the prototype model, the training mechanism, or a mathematical result. The claims in the introduction and conclusion should then be calibrated accordingly.
> >
> > This comparison is necessary for an accurate assessment of the manuscript’s novelty and should go beyond the addition of citations.

---

> ### Author Response · Authors · 2026-07-22
>
> We thank the reviewer for this important comment.  We agree that our previous response did not compare these works directly enough and that the distinction between guarantees for estimators and for specified training dynamics does not distinguish our manuscript from Gavrilopoulos et al. (2024).  We have therefore revised both the comparison and the novelty claims.
>
> Lecué, Li, and Shang (2025) formalize alignment and feature learning through a feature space decomposition.  Their spectral methods can exploit favorable alignment in a fixed representation but do not learn a new representation.  Section 3.2 now clarifies that our alignment examples illustrate this existing distinction.  Gavrilopoulos, Lecué, and Shang (2024) train a representation in the first layer, fix the resulting conjugate kernel, and apply kernel ridge regression to an independent sample. They also relate the learned representation to alignment and excess risk in a single neuron example.  We now discuss this contribution in Sections 2.4.4 and 4.1.
>
> The difference from these works is one of modeling level.  Gavrilopoulos et al. retain a representation derived from a neural network and establish an excess risk result.  Our adaptive feature model is an abstraction in which feature parameters and coefficients evolve jointly, while the over-parameterized Gaussian sequence model removes the neural network architecture and isolates rotations through $A_t$ and rescaling through $D_t$.  The claimed contribution is this prototype model, not a new general definition of feature learning, a training mechanism for neural networks, or a new mathematical result for the prototype.
> Nevertheless, by isolating these mechanisms in a tractable Gaussian sequence setting, the model turns feature adaptation into a concrete statistical object that can be studied beyond fixed feature theories.
>
> Accordingly, we removed the overclaimed “new paradigm”  from the Abstract and Introduction and use "prototype model" instead, and revised the Introduction, Sections 2.4.4, 3.2, 4.1, and 4.3.2, and the Discussion to make these comparisons and boundaries explicit.